# Structural, mechanistic, and physiological insights into phospholipase A-mediated membrane phospholipid degradation in *Pseudomonas aeruginosa*

Florian Bleffert[1], Joachim Granzin[2], Muttalip Caliskan[1], Stephan N Schott-Verdugo[3,4,5], Meike Siebers[6,7], Björn Thiele[8], Laurence Rahme[9], Sebastian Felgner[10], Peter Dörmann[6], Holger Gohlke[2,3,5]*, Renu Batra-Safferling[2]*, Karl-Erich Jaeger[1,11], Filip Kovacic[1]*

[1]Institute of Molecular Enzyme Technology, Heinrich Heine University Düsseldorf, Forschungszentrum Jülich GmbH, Jülich, Germany; [2]Institute of Biological Information Processing - Structural Biochemistry (IBI-7: Structural Biochemistry), Forschungszentrum Jülich GmbH, Jülich, Germany; [3]Institute for Pharmaceutical and Medicinal Chemistry, Heinrich Heine University Düsseldorf, Duesseldorf, Germany; [4]Centro de Bioinformática y Simulación Molecular (CBSM), Faculty of Engineering, University of Talca, Talca, Chile; [5]John von Neumann Institute for Computing (NIC), Jülich Supercomputing Centre (JSC), and Institute of Bio- and Geosciences (IBG-4: Bioinformatics), Forschungszentrum Jülich GmbH, Jülich, Germany; [6]Institute of Molecular Physiology, and Biotechnology of Plants (IMBIO), University of Bonn, Bonn, Germany; [7]Institute for Plant Genetics, Heinrich Heine University Düsseldorf, Düsseldorf, Germany; [8]Institute of Bio- and Geosciences, Plant Sciences (IBG-2), and Agrosphere (IBG-3), Forschungszentrum Jülich GmbH, Jülich, Germany; [9]Department of Microbiology, and Immunobiology, Harvard Medical School, Boston, United States; [10]Department of Molecular Bacteriology, Helmholtz Centre for Infection Research, Braunschweig, Germany; [11]Institute of Bio- and Geosciences (IBG-1: Biotechnology), Forschungszentrum Jülich GmbH, Jülich, Germany

*For correspondence:
gohlke@uni-duesseldorf.de (HG);
r.batra-safferling@fz-juelich.de
(RB-S);
f.kovacic@fz-juelich.de (FK)

Competing interest: The authors declare that no competing interests exist.

**Abstract** Cells steadily adapt their membrane glycerophospholipid (GPL) composition to changing environmental and developmental conditions. While the regulation of membrane homeostasis via GPL synthesis in bacteria has been studied in detail, the mechanisms underlying the controlled degradation of endogenous GPLs remain unknown. Thus far, the function of intracellular phospholipases A (PLAs) in GPL remodeling (Lands cycle) in bacteria is not clearly established. Here, we identified the first cytoplasmic membrane-bound phospholipase $A_1$ (PlaF) from *Pseudomonas aeruginosa*, which might be involved in the Lands cycle. PlaF is an important virulence factor, as the *P. aeruginosa* ΔplaF mutant showed strongly attenuated virulence in *Galleria mellonella* and macrophages. We present a 2.0-Å-resolution crystal structure of PlaF, the first structure that reveals homodimerization of a single-pass transmembrane (TM) full-length protein. PlaF dimerization, mediated solely through the intermolecular interactions of TM and juxtamembrane regions, inhibits its activity. The dimerization site and the catalytic sites are linked by an intricate ligand-mediated interaction network, which might explain the product (fatty acid) feedback inhibition observed with the purified PlaF protein. We used molecular dynamics simulations and configurational free energy computations to suggest a model of PlaF activation through a coupled monomerization and tilting of the monomer in the membrane, which constrains the active site cavity into contact with the GPL

substrates. Thus, these data show the importance of the PlaF-mediated GPL remodeling pathway for virulence and could pave the way for the development of novel therapeutics targeting PlaF.

## Editor's evaluation

This study provides new insights into how a bacterial phospholipase called PlaF degrades membrane phospholipids in a controlled fashion to allow bacteria to alter their membrane composition to adapt to changing conditions. Inas much as PlaF is important for virulence, it will be interesting to see if the comprehensive biochemical and structural analysis in the current paper will aid in the development of a class of antibiotics targeting PlaF.

## Introduction

Biological membranes are steadily changing and adapting to environmental and developmental conditions (*Eickhoff and Bassler, 2018*; *Parsons and Rock, 2013*). These changes affect processes indispensable for cell life, such as nutrient uptake (*Higgins, 1992*), chemical signaling (*Venturi and Fuqua, 2013*), protein secretion (*Krampen et al., 2018*), folding (*Mackenzie, 2006*), interaction with hosts (*Baxter et al., 2015*), and antibiotic resistance (*García-Fernández et al., 2017*). An important mechanism to maintain membrane functionality in bacteria is the alteration of lipid composition (*Rowlett et al., 2017*; *Schniederjans et al., 2017*; *Zhang and Rock, 2008*). The adjustment of the fatty acid (FA) composition of glycerophospholipids (GPLs) upon thermal adaptation represents one of the most important mechanisms of membrane lipid homeostasis (*Sinensky, 1974*; *Cossins, 1994*). Adaptive changes in membrane GPL composition were observed under numerous other conditions, including environmental stresses (*Rowlett et al., 2017*), the transition from planktonic to sessile lifestyle (*Benamara et al., 2014*), and heterologous protein production (*Kanonenberg et al., 2019*).

De novo synthesis of GPLs is the main pathway used to tune the proportions of different lipid classes in bacteria (*Zhang and Rock, 2008*; *Jeucken et al., 2019*). Furthermore, bacteria rapidly alter their membrane GPL composition by chemical modifications (cis-trans isomerization and cyclopropanation) of acyl chains in GPLs to respond to environmental changes (*Zhang and Rock, 2008*). However, the bacterial pathway for remodeling of GPLs involving a rapid turnover of the acyl chains of GPLs is unknown. Interestingly, such a pathway was discovered in eukaryotes by W. E. Lands more than 60 years ago (*Lands, 1958*). This Lands cycle involves PLA-catalyzed deacylation of membrane GPLs to mono-acyl GPLs (lysoGPLs) followed by lysophospholipid acyltransferase (LPLAT)-mediated acylation of lysoGPL to yield a new GPL molecule with acyl chain composition different from the starting GPL (*Lands, 1958*). Despite the importance of this metabolic process in different animal and plant tissues, it took nearly 50 years before the enzymes involved in phospholipid remodeling were discovered (*Shindou and Shimizu, 2009*). Fourteen different mammalian LPLAT with specificities for different GPL head groups were reported to be involved in the Lands cycle (*Hishikawa et al., 2008*; *Valentine et al., 2022*). The recently published structure of human LPLAT provided the first insights into the molecular mechanism by which lysoGPL is acylated to GPL (*Zhang et al., 2021*). At least 16 mammalian PLAs (cytosolic and calcium-independent families) that may act on the membrane GPLs with different substrate profiles and tissue expression patterns are known (*Clark et al., 1990*; *Song et al., 1999*; *Underwood et al., 1998*; *Ohto et al., 2005*). Some PLAs have a suggested role in the remodeling of membrane GPLs (*Asai et al., 2003*), while others are involved in producing lipid mediators and bioenergetics (*Murakami et al., 2020*). Detailed computational studies revealed that human iPLA$_2\beta$ is allosterically activated by binding to the membrane, which is required to extract a single GPL molecule from the membrane and subsequent hydrolysis (*Mouchlis et al., 2015*).

Whereas extensive studies have been carried out for secreted bacterial PLAs acting as host-cell effectors (*Istivan and Coloe, 2006*), only limited information is available for the enzymes from the intracellular PLA family (*Flores-Díaz et al., 2016*). Previously, we reported that periplasmic TesA from *Pseudomonas aeruginosa* is a multifunctional enzyme with lysoPLA activity (*Kovačić et al., 2013*). However, this enzyme has no PLA activity, and therefore it is most likely not related to membrane GPL remodeling (*Leščić Ašler et al., 2010*). We recently published a novel intracellular PLA from *P. aeruginosa* whose function for remodeling of GPLs still needs to be experimentally analyzed (*Weiler et al., 2022*). Comprehensive lipidomic profiling of 113 *Escherichia coli* strains with deleted or overexpressed

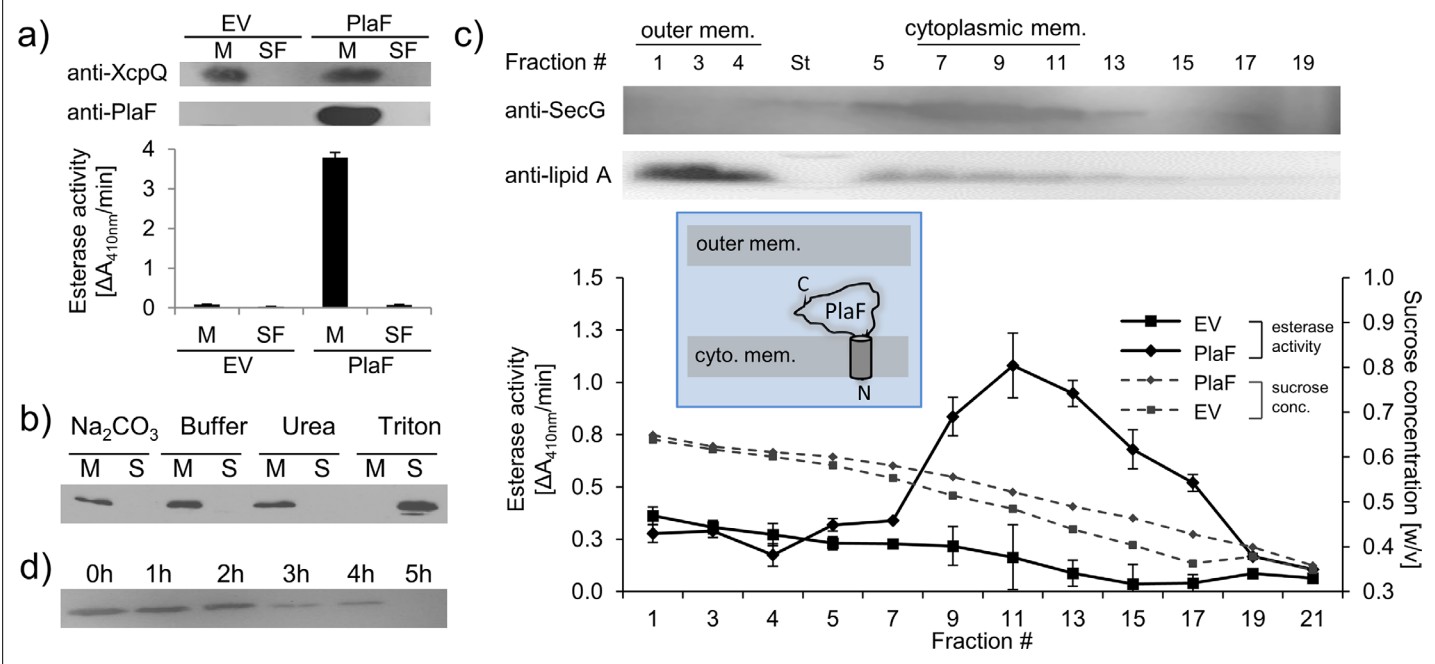

**Figure 1.** Subcellular localization of PlaF. (**a**) PlaF is a membrane protein of *Pseudomonas aeruginosa*. The membrane (M) and soluble fractions (SFs) of cell extracts from *P. aeruginosa* p-*plaF*, and the empty vector control strain (EV) were separated, analyzed by immunodetection with anti-His$_6$-tag antibodies, and by esterase activity assay. The membrane protein marker *P. aeruginosa* XcpQ was detected with anti-XcpQ antibodies. (**b**) PlaF is an integral membrane protein of *P. aeruginosa*. The crude membranes of *P. aeruginosa* p-*plaF* were treated with sodium carbonate, urea, Triton X-100, or MES buffer control followed by ultracentrifugation (S, supernatant; M, membrane proteins). PlaF was detected as in (**a**). (**c**) PlaF is a cytoplasmic-membrane protein of *P. aeruginosa*. The membrane fractions of *P. aeruginosa* p-*plaF* and the EV strains were isolated and separated by ultracentrifugation in a sucrose density gradient. The esterase activity was assayed as in (**a**). *P. aeruginosa* SecG, and outer membrane lipid A were used as markers for cytoplasmic, and outer membranes, and detected by Western blotting using anti-SecG, and anti-Lipid A antibodies, respectively. Inlet: A model of PlaF cellular localization. All values are mean ± standard deviation (S.D.) of three independent experiments measured in triplicates. (**d**) The catalytic domain of PlaF is exposed to the periplasm. *P. aeruginosa* p-*plaF* cells with permeabilized outer membrane were treated with trypsin for the indicated periods, and PlaF was detected as described in (**a**).

The online version of this article includes the following source data for figure 1:

**Source data 1.** Uncropped Western blot shown in *Figure 1a*.

**Source data 2.** Uncropped Western blot shown in *Figure 1b*.

**Source data 3.** Uncropped Western blot shown in *Figure 1c*.

**Source data 4.** Uncropped Western blot shown in *Figure 1d*.

**Source data 5.** Excel file with data used to make *Figure 1a*.

**Source data 6.** Excel file with data used to make *Figure 1c*.

lipid metabolism genes did not reveal the identity of an intracellular PLA suitable for the Lands cycle (*Jeucken et al., 2019*). Here, we describe PlaF from *P. aeruginosa* (*Kovacic et al., 2016*; *Bleffert et al., 2019*) as the first cytoplasmic membrane-bound PLA with a role in virulence and GPL remodeling pathway in bacteria. We determined the crystal structure of PlaF (*Kovacic et al., 2016*; *Bleffert et al., 2019*) as a basis to provide mechanistic insights into PLA-mediated membrane phospholipid degradation related to bacterial virulence.

## Results

### PlaF is an integral cytoplasmic membrane-bound enzyme

We previously purified PlaF from the Triton X-100 solubilized membranes of a *P. aeruginosa* strain carrying the p-*plaF* expression plasmid (*Kovacic et al., 2016*; *Bleffert et al., 2019*). Here, we show that catalytically active PlaF is an intrinsic integral membrane protein as it was absent in the soluble fraction of the *P. aeruginosa* p-*plaF* (*Figure 1a*) and remained membrane-associated after treatment of

PlaF-containing membranes with denaturation agents ($Na_2CO_3$ or urea), which destabilize weak interactions between peripheral proteins and the membrane (*Figure 1b*). To identify if PlaF is associated with the inner or outer membrane, *P. aeruginosa* p-*plaF* membranes were fractioned by ultracentrifugation in a sucrose density gradient. Western blot analysis of the cytoplasmic membrane protein SecG (*Bleves et al., 1996*), and the outer membrane-associated Lipid A (*Matsushita et al., 1978*) combined with PlaF activity measurement revealed that the majority of PlaF was in the cytoplasmic membrane fractions (#9–13) (*Figure 1c*). As expected, the Lipid-A-containing fractions (#1–3) showed negligible PlaF activity (*Figure 1c*), overall demonstrating that PlaF is a cytoplasmic integral membrane protein. Proteolysis experiments in which *P. aeruginosa* p-*plaF* cells with a chemically permeabilized outer membrane were treated with trypsin revealed a time-dependent degradation of PlaF (*Figure 1d*). These results suggest that PlaF is likely anchored to the cytoplasmic membrane via a transmembrane (TM) domain at the N-terminus predicted from sequence analysis (*Kovacic et al., 2016*), and its catalytic C-terminal domain protrudes into the periplasm.

## PlaF is a PLA₁ involved in the alteration of membrane GPL composition as determined by global lipidomics

The previously reported carboxylesterase activity of PlaF (*Bleffert et al., 2019*) was here further analyzed using different PLA substrates. PlaF, purified with *n*-octyl-β-D-glucoside (OG) as described previously (*Kovacic et al., 2016*), showed PLA₁ but no PLA₂ activity toward the artificial substrates specific to each of these two phospholipase families (*Figure 2a*) and the natural phospholipid diacyl phosphatidylglycerol containing pentanoic and oleic acid at the *sn*-1 and *sn*-2 positions, respectively (*Figure 2—figure supplement 1*). The substrate profile of PlaF against natural di-acyl GPLs commonly occurring in *P. aeruginosa* membranes (*Benamara et al., 2014*) was determined with a spectrum of substrates (see legend to *Figure 2b*). In vitro, purified PlaF preferably hydrolyzed GPLs containing medium-chain FAs (C12 and C14) and showed comparable activities with phosphatidylethanolamine (PE), phosphatidylglycerol (PG), and phosphatidylcholine (PC) (*Figure 2b*).

To examine the role of membrane-bound PlaF in the regulation of the membrane GPL composition in vivo, we constructed the *P. aeruginosa* deletion mutant Δ*plaF* lacking the entire *plaF* gene by homologous recombination, and a complemented Δ*plaF::plaF* strain as a control (*Figure 2—figure supplement 2*). The activity assay showed ~90% loss of PLA₁ activity in the mutant strain, and restoration of activity in Δ*plaF::plaF* slightly above the wild-type (WT) level (*Figure 2—figure supplement 2*). These findings indicate that PlaF is a major but not the only intracellular PLA₁ in *P. aeruginosa*.

The quantitative mass spectrometric (Q-TOF-MS/MS) analysis of total GPLs isolated from four biological replicates of *P. aeruginosa* WT, Δ*plaF*, and Δ*plaF::plaF* cells revealed significant differences in membrane GPL composition (*Figure 2c*, *Supplementary files 1-3*). Statistical analysis of 323 GPL molecular species identified six significantly ($p<0.05$) accumulating GPLs, varying in the composition of head groups (PE, PG, PC, and phosphatidylinositol, PI), length, and unsaturation of acyl chains, in *P. aeruginosa* Δ*plaF*. Interestingly, these GPLs were present at low concentrations in the cells which may explain why they were not detected in the previous lipidomic analyses of *P. aeruginosa* GPLs (*Benamara et al., 2014*; *Le Sénéchal et al., 2019*). In the complemented strain (Δ*plaF::plaF*), these GPLs were depleted compared to the Δ*plaF*, although not to the WT level (*Supplementary file 2*). These results strongly indicate that PlaF specifically hydrolyses low abundant GPLs in vivo. We furthermore observed that the other seven PE, PG, and PC species, which belong among the most abundant *P. aeruginosa* GPLs (*Benamara et al., 2014*; *Le Sénéchal et al., 2019*), were significantly depleted (*Figure 2c*) in *P. aeruginosa* Δ*plaF*, and their concentrations were significantly elevated in complementation strain (*Figure 2c*). This may explain why the net GPL contents of the WT and the Δ*plaF* strain were not significantly ($p=0.67$) different. Significantly affected GPLs in the Δ*plaF* strain account for ~11% (mol/mol) of the total GPL content, indicating the profound function of PlaF in membrane GPL remodeling.

Our quantitative lipidomics results, which revealed that several PE, PG, and PC molecular species accumulated or were depleted in Δ*plaF*, together with in vitro PLA activity data of PlaF with various PE, PG, and PC substrates, indicate that PlaF might be a major PLA involved in the Lands cycle (*Figure 2d*). Thus, the six low-abundant PE, PG, and PC species that accumulated in Δ*plaF* might be PlaF substrates. PlaF-mediated hydrolysis of these substrates yields lysoGPL intermediates. Acylation

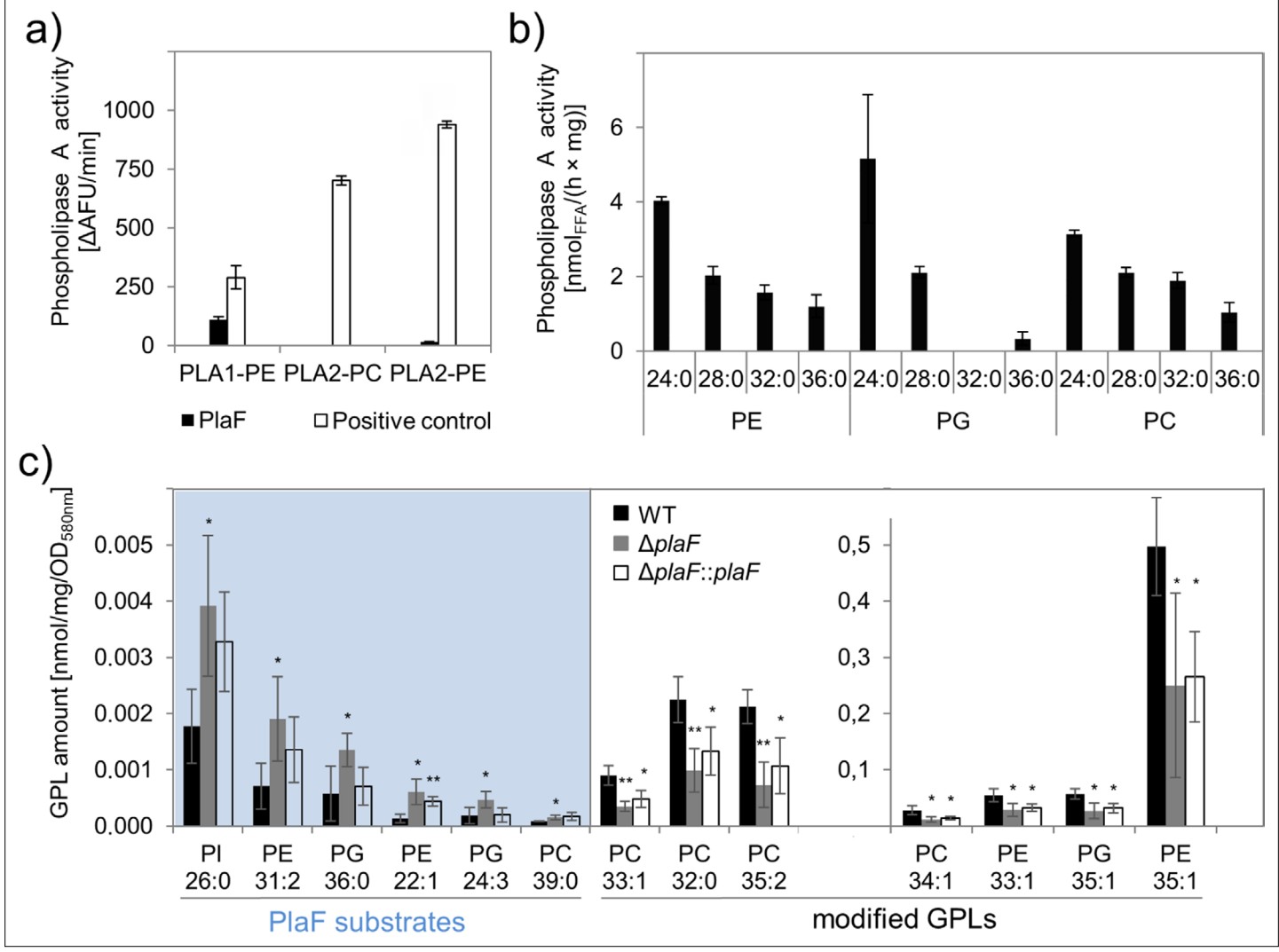

**Figure 2.** Phospholipolytic activity profiling of PlaF. (**a**) PlaF is a phospholipase $A_1$. Enzyme activities of PlaF were measured fluorimetrically using artificial $PLA_1$, and $PLA_2$ substrates containing either ethanolamine (PE) or choline (PC) head groups. The control enzymes were $PLA_1$ of *Thermomyces lanuginosus,* and $PLA_2$ of *Naja mocambique*. Results are means±S.D. of three independent measurements performed with at least three samples. (**b**) PlaF releases FAs from naturally occurring bacterial GPLs. PLA activity of PlaF was measured by quantification of released FAs after incubation of PE, PG, and PC substrates containing FAs with different chain lengths (C12–C18). (**c**) PlaF changes GPL composition of *Pseudomonas aeruginosa* cells. Crude lipids extracted from *P. aeruginosa* wild-type (WT), Δ*plaF*, and Δ*plaF::plaF* membranes were quantified by Q-TOF-MS/MS using an internal standard mixture of GPLs. PlaF substrates are elevated in Δ*plaF* and depleted in Δ*plaF::plaF*, while modified GPLs show inverse response than GPL substrates. The GPL amount (nmol) was normalized to mg of crude lipids, and optical density (***Supplementary file 3***). FA composition of GPL is depicted as XX:Y, where XX defines the number of carbon atoms, and Y defines the number of double bonds in FAs bound to GPL. Results are mean ± S.D. of four biological replicates of WT, Δ*plaF, and* three of the Δ*plaF::plaF*. T-test of normally distributed values, ** $p<0.01$, * $p<0.05$. FA, fatty acid; GPL, glycerophospholipid.

The online version of this article includes the following source data and figure supplement(s) for figure 2:

**Source data 1.** Excel file with data used to make ***Figure 2a***.

**Source data 2.** Excel file with data used to make ***Figure 2b***.

**Figure supplement 1.** Determination of PLA activity of PlaF by GC-MS.

**Figure supplement 1—source data 1.** Excel file with data used to make ***Figure 2—figure supplement 1***.

**Figure supplement 2.** Generation of *Pseudomonas aeruginosa* ΔplaF deletion mutant and *P. aeruginosa* ΔplaF::plaF complemented strain.

**Figure supplement 2—source data 1.** Excel file with data used to make ***Figure 2—figure supplement 2***.

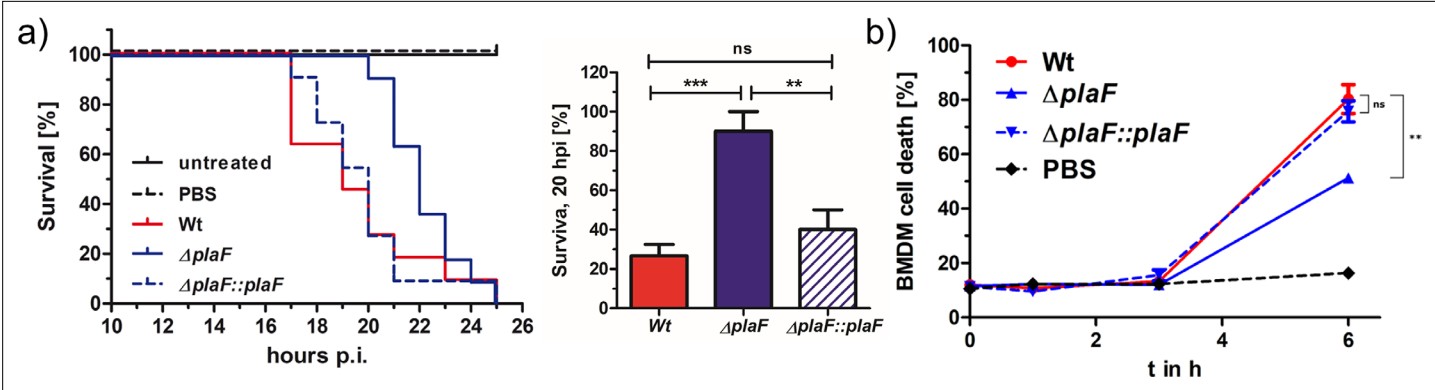

**Figure 3.** PlaF is a novel virulence factor of *Pseudomonas aeruginosa* PAO1. (**a**) Left: *P. aeruginosa* Δ*plaF* strain is less virulent than the respective wild-type (WT) strain in a *Galleria mellonella* larvae virulence assay. Kaplan-Meier plot of representative data of at least two experiments with 10 larvae per group. PBS treated and untreated larvae served as infection and viability controls, respectively. Right: Statistical analysis of the survival at the 20 hr using three independent experiments with 10 larvae each. (**b**) *P. aeruginosa* Δ*plaF* strain is less cytotoxic to bone marrow-derived macrophages (BMDMs) than the WT strain in cell culture. The BMDM cells ($5×10^5$) were infected with $5×10^5$ bacteria in a 24-well plate, and lactate dehydrogenase activity in supernatants was determined as a measure of BMDM death. The Δ*plaF* phenotype could be complemented with *P. aeruginosa* Δ*plaF::plaF*. PBS or Triton-X100 (1% v/v) treated cells served as viability or 100% killing controls, respectively. Results are the representative data of two independent experiments (n=10). One-way ANOVA analysis, *** $p<0.001$, ** $p<0.01$, ns, not significant; PBS, phosphate-buffered saline.

The online version of this article includes the following source data and figure supplement(s) for figure 3:

**Figure supplement 1.** The growth of *Pseudomonas aeruginosa* PA01 and ΔplaF do not differ.

**Figure supplement 1—source data 1.** Excel file with data used to make *Figure 3—figure supplement 1*.

**Figure supplement 2.** Sequence alignment of PlaF and its homologs.

of these lysoGPLs by an unknown acyltransferase will produce modified GPLs typical to *P. aeruginosa*. The absence of lysoGPL intermediates in Δ*plaF* will lead to the depletion of modified GPLs (*Figure 2d*).

## PlaF is a potent virulence factor of *P. aeruginosa* affecting in vivo toxicology

We next addressed the question of whether PlaF contributes to the virulence of *P. aeruginosa* by using the *G. mellonella* infection model and the bone marrow-derived macrophages (BMDMs) viability assay. The results revealed a remarkable difference in the survival of *G. mellonella* larvae infected with *P. aeruginosa* WT or Δ*plaF*. While Δ*plaF* was avirulent during 20 hr of infection, the majority of the larvae (~80 %) did not survive 20 hr after infection with the *P. aeruginosa* WT (*Figure 3a*). The viability assays with *P. aeruginosa*-infected BMDMs showed a significantly ($p<0.01$) stronger killing effect of *P. aeruginosa* WT compared to Δ*plaF* 6 hr after infection (*Figure 3b*). As expected, the complemented strain (Δ*plaF::plaF*) restored the loss of virulence of Δ*plaF* in *G. mellonella, and* BMDM assays (*Figure 3a and b*). Comparison of the growth curves of *P. aeruginosa* Δ*plaF*, and the WT in nutrient-rich medium (*Figure 3—figure supplement 1*) showed that PlaF most likely does not reduce virulence by affecting the growth of *P. aeruginosa*.

A BLAST search revealed PlaF orthologs in more than 90% of all sequenced *P. aeruginosa* genomes, including 571 clinical isolates (*Supplementary file 4*). Furthermore, we found PlaF homologs in pathogens from the *Pseudomonas* genus (*P. alcaligenes, P. mendocina, and P. otitidis*), and other high-priority pathogens (*Acinetobacter baumannii*, *Klebsiella pneumoniae*, and *Streptococcus pneumoniae*) (*Figure 3—figure supplement 2*). These results indicate that PlaF is a novel and potent *P. aeruginosa* virulence factor, which is conserved in important pathogens and, therefore, might be a promising target for developing novel broad-range antibiotics.

**Table 1.** Data collection and refinement statistics on PlaF.

| X-ray data | |
|---|---|
| Beamline/detector | ID29, ESRF (Grenoble, France)/DECTRIS PILATUS 6M |
| Wavelength (Å)/monochromator | $\lambda$ =0.96863/channel-cut silicon monochromator, Si (111) |
| Resolution range (Å) | 47.33–2.0 (2.05–2.0)* |
| Space group | I 4₁ 2 2 |
| Unit cell (a=b), c (Å); α=β=γ | a=133.87 c=212.36; 90° |
| Total reflections | 669,964 (47,385) |
| Unique reflections | 65,113 (4527) |
| Multiplicity | 10.3 (10.5) |
| Completeness (%) | 100.0 (100.0) |
| Mean I/sigma (I) | 24.6 (2.5) |
| Wilson B-factor (Å²) | 38.3 |
| R-merge (%) | 5.3 (91.3) |
| R-meas (%) | 5.6 (100.6) |
| **Refinement** | |
| R-work (%) | 16.3 (23.15) (2.071–2.0)* |
| R-free (%) | 18.57 (27.81) |
| Number of atoms | 5187 |
| Macromolecules | 4831 |
| Ligands | 123 |
| Water | 233 |
| Protein residues | 620 |
| RMS (bonds) | 0.008 |
| RMS (angles) | 1.07 |
| Ramachandran favored (%) | 99 |
| Ramachandran outliers (%) | 0 |
| Clashscore | 3.14 |
| Average B-factor (Å²) | 49.1 |
| Macromolecules (Å²) | 48.8 |
| Ligands (Å²) | 79.2 |
| Solvent (Å²) | 47.9 |

*Values in parentheses are for the highest resolution shell.

## Crystal structure of PlaF homodimer in the complex with natural ligands

To gain insights into the PlaF structure, we crystallized the OG-solubilized PlaF protein purified from *P. aeruginosa* membranes as described previously (*Bleffert et al., 2019*). The structure was refined at a resolution of up to 2.0 Å (*Table 1*). The final model in the asymmetric unit consists of two protein molecules (PlaF_A and PlaF_B), which are related by improper twofold non-crystallographic symmetry (*Figure 4a*). Active site cavities of both the monomers reveal non-covalently bound ligands—myristic acid (MYR), OG, and isopropyl alcohol (IPA) in PlaF_A; and undecyclic acid (11A), OG, and IPA in PlaF_B (*Figure 4a*, *Supplementary file 5*). These FAs are the natural ligands from the homologous organism

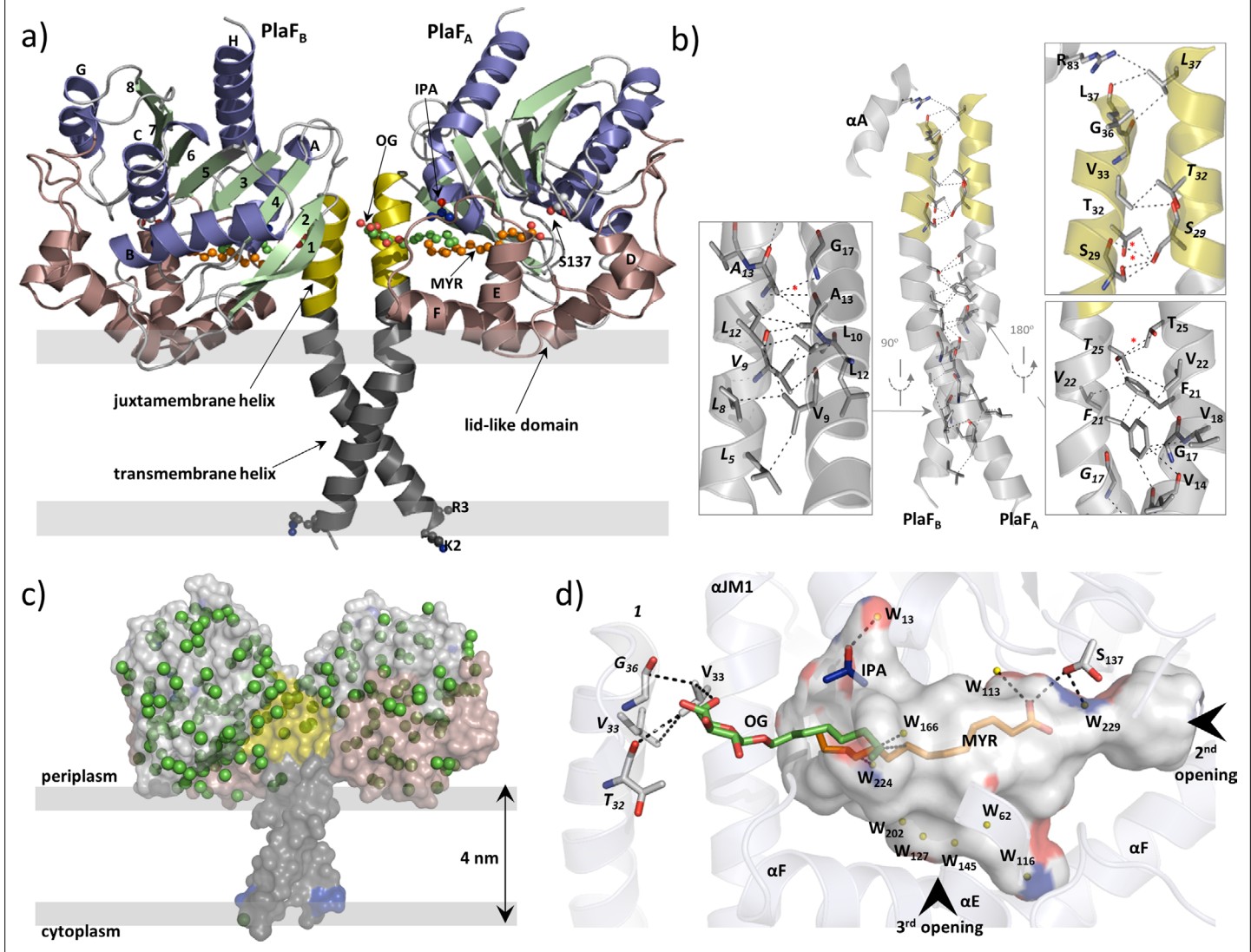

**Figure 4.** Overall structure of dimeric PlaF with bound endogenous FA ligands. (**a**) A unique N-terminal helix comprising a putative transmembrane helix (αTM1, $L_5$–$L_{27}$, gray) flanked by charged residues ($K_2$, $R_3$) on one side and, on another side, the juxtamembrane helix (αJM1, $A_{28}$–$L_{37}$, yellow). αJM1 links the αTM1 with the catalytic domain, which consists of an α/β-hydrolase (blue, α-helices; green, β-strands, and gray, loops), and a lid-like domain (brown). Ligands bound in the active site cleft are shown as ball-and-sticks (oxygen, red; carbon of OG, MYR, and IPA, green, orange, and blue, respectively). Thick gray lines roughly depict the membrane borders. (**b**) Dimer interface. Interactions involving TM-JM helices are predominantly hydrophobic with four weak H-bonds (indicated by a red asterisk) detected mostly in the αJM1. $R_{83}$ is the only residue outside of the JM-TM helix involved in interactions. Residues of the PlaF_B molecule are indicated in italics. A detailed list of interactions is provided in ***Supplementary file 6***. (**c**) A model suggesting the orientation of PlaF in the membrane. The water molecules are indicated as green spheres. The transparent surface of PlaF was colored as in (**a**). PlaF is rotated by 180° along the normal to the membrane compared with ***Figure 4***. (**d**) Interaction network within the ligand-binding cleft of PlaF_A. MYR is linked via H-bond with the catalytic $S_{137}$, and via hydrophobic interactions with OG. The sugar moiety of OG from PlaF_A forms H-bonds with $V_{33}$ of PlaF_A, which is interacting with $V_{33}$ and $G_{36}$ of PlaF_B. The part of the cleft in the direction of the opening 3 is occupied by several water molecules (W, yellow spheres). The cleft accommodates one IPA molecule bound to the water. Arrows indicate two openings not visible in this orientation. The cleft was calculated using the Pymol software and colored by elements: carbon, gray; oxygen, red; nitrogen, blue. FA, fatty acid.

The online version of this article includes the following figure supplement(s) for figure 4:

**Figure supplement 1.** Identification of fatty acid (FA) ligands co-purified with PlaF.

**Figure supplement 2.** Comparison of PlaF monomers.

**Figure supplement 3.** TM-JM helix of PlaF is not detected among PlaF structural homologs.

**Figure supplement 4.** The lid-like domains of PlaF and its homologs.

**Figure supplement 5.** PlaF structure reveals differently ordered subdomains.

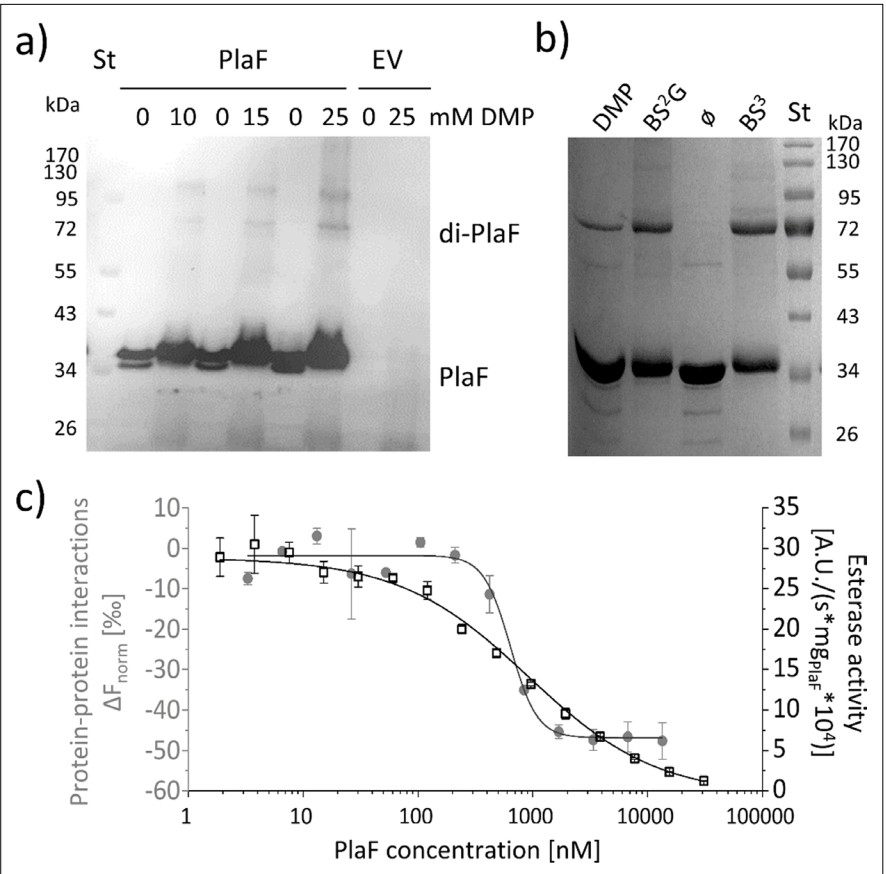

**Figure 5.** PlaF oligomeric states and their enzymatic activity. (**a**) PlaF forms dimers in cell membranes.In vivo cross-linking experiments were performed by incubating *Pseudomonas aeruginosa* p-*plaF* or the empty vector control (EV) cells with different concentrations of DMP cross-linker followed by immunodetection of PlaF with anti-PlaF antiserum. (**b**) In vitro cross-linking of purified PlaF. Purified PlaF was incubated with DMP, BS$^2$G, and BS$^3$ cross-linking reagents or buffer control (ø) for 90 min, and the samples were analyzed by SDS-PAGE. Molecular weights of protein standard in kDa are indicated. (**c**) PlaF homodimerization, and activity are concentration-dependent. Protein-protein interactions of purified PlaF were monitored by measuring the changes in thermophoresis ($\Delta F_{norm}$, gray circles) using the MST method. The MST results are mean ± S.D. of two independent experiments with PlaF purified with OG. Esterase activity (black squares) of PlaF was measured in three independent experiments using 4-methylumbelliferyl palmitate substrate. Dissociation ($K_D$) and activation ($K_{act}$) constants were calculated using a logistic fit of sigmoidal curves.

The online version of this article includes the following source data and figure supplement(s) for figure 5:

**Source data 1.** Uncropped Western blot shown in *Figure 5a*.

**Source data 2.** Uncropped SDS-PAGE shown in *Figure 5b*.

**Source data 3.** Origin file with data used to make *Figure 5c*.

**Figure supplement 1.** In vivo crosslinking of PlaF.

**Figure supplement 1—source data 1.** Uncropped SDS-PAGE shown in *Figure 5—figure supplement 1*.

**Figure supplement 1—source data 2.** Original file of the SDS-PAGE shown in *Figure 5—figure supplement 1*.

**Figure supplement 2.** Size exclusion chromatography of PlaF showed a monomer.

*P. aeruginosa* that were co-purified with PlaF, as confirmed by gas chromatography-mass spectrometric (GC-MS) analysis of organic solvent extracts of purified PlaF (*Figure 4—figure supplement 1*). Compared to the protein chains, the bound FAs have higher average B-factor values for 11A (89.0 Å$^2$) and MYR (66.6 Å$^2$), indicating different flexibility of the ligands bound to the active sites of the two PlaF molecules.

The N-terminal 38 amino acids form a long, kinked helix that comprises the putative TM (αTM1) and the JM (αJM1) helices, connecting the catalytic domain with the membrane (*Figure 5a*). The kink angle in the TM-JM helices is the main difference between the two monomers (*Figure 4—figure supplement 2*) and is likely caused by crystal packing effects (*Figure 4—figure supplement 2*). Dimerization is mediated primarily via hydrophobic interactions between symmetry-nonrelated residues from the TM-JM domains of two monomers (*Figure 4b*, *Supplementary file 6*), consistent with the hydrophobic effects that dominate in the stabilization of dimeric TM domains (*MacKenzie et al., 1997*). In addition, four weak H-bonds (*Figure 4b*) between JM residues stabilize the PlaF dimer. The TM-JM helices adopt a coiled-coil-like conformation (*Figure 4—figure supplement 2*), where the αTM1 crosses its counterpart at V14 to form an elongated X-shaped dimer interface with the buried surface area of 656 Å$^2$ per monomer. The full-length PlaF dimer represents a unique structure, as neither a relevant match to the TM-JM helix (*Figure 4—figure supplement 3*) nor the membrane-spanning coiled-coil structure of the TM-JM dimer has been reported previously.

## The crystal structure of PlaF is indicative of a specific orientation in the membrane

The catalytic domain of PlaF adopts a canonical α/β-hydrolase fold (*Ollis et al., 1992*; *Figure 4a*) with three α-helices forming a distinct lid-like domain that covers the active site (*Figure 4a*). Despite the high homology of the catalytic domain, the lid-like domain varies significantly between PlaF homologs (*Figure 4—figure supplement 4*), as observed previously for other lipolytic enzymes (*Figure 4—figure supplement 4*; *Chow et al., 2012*). Furthermore, the lid-like domain shows a less ordered structure, as suggested by comparatively higher B-factors (*Figure 4—figure supplement 5*). This is likely a consequence of the lack of stabilizing interactions between the charged residue-rich (23 of the 77 residues) lid-like domain and the hydrophilic head groups of membrane GPLs in the native membrane environment. The TLS (translation-libration-screw-rotation) model revealed higher disorder in the TM-JM domains, presumably also due to the missing interactions with the hydrophobic membrane (*Figure 4—figure supplement 5*). No ordered water molecules in the vicinity of αTM1 (*Figure 4c*) and the presence of several charged and polar residues adjacent to αTM1 suggest a model where the TM-JM domain spans through the membrane with the catalytic domain localized on the membrane surface (*Figure 4c*).

## Ligand-mediated interaction network connects dimerization and active sites

The active site of PlaF comprises the typical serine-hydrolase catalytic triad with $S_{137}$, $D_{258}$, and $H_{286}$ interacting through H-bonds (*Jaeger et al., 1994*; *Supplementary file 7*). Interestingly, $S_{137}$ shows two side-chain conformations, where one conformer is within the hydrogen bond distance of the FA ligand (*Figure 4d*, *Supplementary files 5 and 7*). Additionally, $S_{137}$ forms H-bonds with residues $I_{160}$, $D_{161}$, and $A_{163}$ located in the lid-like domain. The active site cleft in PlaF is formed by residues from the helix αJM1, the α/β-hydrolase and the lid-like domains (*Figure 4d*, *Supplementary file 8*). In PlaF, the large T-shaped active site cleft formed by residues from the JM helix, the α/β-hydrolase, and the lid-like domains is amphiphilic, making it compatible with binding the bulky GPL substrates. Three openings are observed in the cleft—one, close to the catalytic $S_{137}$, lined with residues from the loops preceding αE, and αF; second, in the middle pointing toward the putative membrane, lined mostly with polar residues of the loops preceding αB, and αF; and third, at the dimer interface, comprising residues from αJM1, and the loop preceding αF of the lid-like domain. The third opening accommodates a pseudo-ligand OG (*Figure 4d*), which with its pyranose ring interacts with residue $V_{33}$ of PlaF$_A$, which in turn participates in dimerization via interactions with $V_{33}$ and $T_{32}$ of PlaF$_B$ (*Figure 4b*). The alkyl chains of OG and MYR bound in the active site cleft are stabilized via hydrophobic interactions (*Figure 4d*). Finally, the H-bond interaction of catalytic $S_{137}$ with the carboxyl group of MYR completes an intricate ligand-mediated interaction network bridging the catalytic ($S_{137}$) and dimerization ($V_{33}$) sites in PlaF (*Figure 4d*). The crystal structure presented thus suggests a role of dimerization and ligand binding in regulating PlaF function, which was subsequently analyzed biochemically.

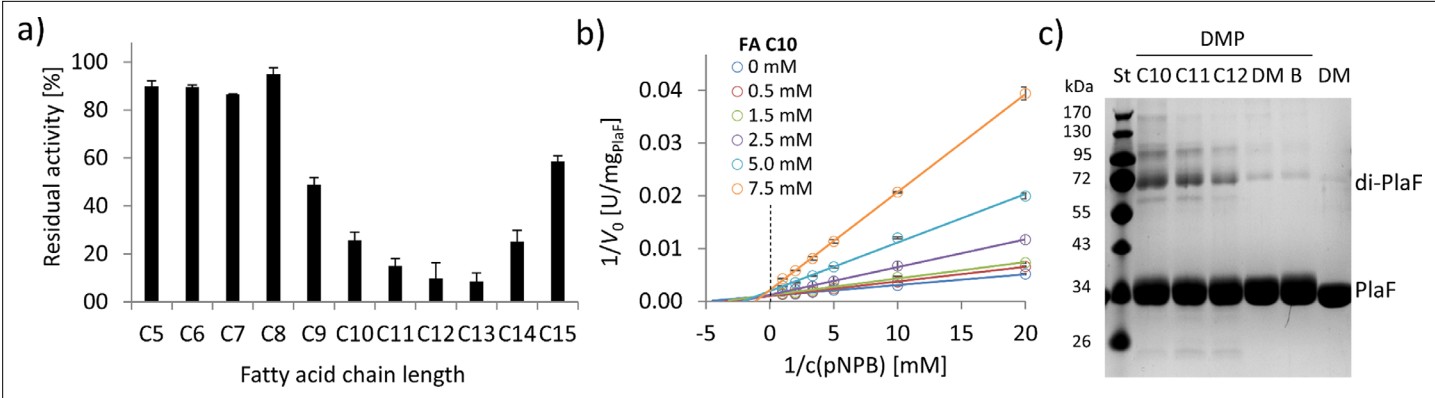

**Figure 6.** FAs exert an inhibitory effect on PlaF and trigger dimerization. (**a**) Inhibition of PlaF with FAs. Esterase activity of PlaF was measured in the presence of 7.5 mM FA (C5–C15); an untreated PlaF sample was set as 100%. The results are mean ± S.D. of three experiments with three samples each. (**b**) Kinetic studies with FA C10 show evidence of mixed-inhibition. Double-reciprocal plots of initial reaction velocities measured with the *p*-NPB substrate and FA C10 inhibitor at concentrations in a range of 0–7.5 mM. (**c**) The effect of FAs on PlaF dimerization. PlaF samples incubated with FAs (C10–C12), dimethyl sulfoxide (DM, DMSO used to dissolve FAs), and purification buffer (B, dilution control) were cross-linked with dimethyl pimelimidate (DMP). FA, fatty acid.

The online version of this article includes the following source data for figure 6:

**Source data 1.** Excel file with data used to make *Figure 6a*.

**Source data 2.** Uncropped SDS-PAGE shown in *Figure 6c*.

**Source data 3.** Excel file with data used to make *Figure 6b*.

## The PlaF activity is affected by dimerization

To investigate the oligomeric state of PlaF in vivo, we performed cross-linking experiments in which intact *P. aeruginosa* p-*plaF* cells were incubated with the cell-permeable bi-functional cross-linking reagent dimethyl pimelimidate (DMP). Western blot results revealed the presence of monomeric and dimeric PlaF in DMP-treated cells, whereas dimers were absent in untreated cells (*Figure 5a* and *Figure 5—figure supplement 1*). Size exclusion chromatography showed that PlaF was extracted from the membranes with detergent and purified by IMAC elutes as a monomer (*Figure 5—figure supplement 2*). Incubation of this purified PlaF for 90 min with bi-functional cross-linkers of different lengths (DMP; bis(sulfosuccinimidyl) glutarate, BS²G or bis(sulfosuccinimidyl) suberate, BS³) resulted in the formation of a substantial amount of PlaF dimers, suggesting spontaneous dimerization in the solution (*Figure 5b*). Microscale thermophoresis (MST) measurements were performed in which the fluorescence-labeled PlaF was titrated with an equimolar concentration of non-labeled PlaF to quantify spontaneous dimerization. The results revealed a sigmoidal binding curve from which a dissociation constant $K_D=637.9\pm109.4$ nM was calculated, indicating weak binding (*Figure 5c*). Measurements of the esterase activity of PlaF samples used for MST experiments revealed that the specific activity of PlaF strongly decreased with increasing PlaF concentrations (*Figure 5c*). Enzyme activity measurements were employed to calculate the activation constant $K_{act}=916.9\pm72.4$ nM. The similar dissociation and activation constants support a model in which PlaF activity is regulated through reversible dimerization in vitro.

## FAs induce dimerization and inhibit PlaF

To investigate the effect of FA ligands on the activity of PlaF, we used mM concentrations of FAs with different chain lengths (C5–C15) in a competitive inhibition assay. PlaF was strongly inhibited (>80%) with FAs containing 10–14 carbon atoms (*Figure 6a*), while the shorter and longer FAs showed only moderate to weak inhibition (*Figure 6a*). To explore the underlining mechanism, we performed kinetic inhibition studies with increasing concentrations of decanoic acid (C10). The results showed that C10 FA lowered maximal hydrolysis rates ($v_{max}$) as expected for a competitive inhibitor. Yet, elevated binding constants ($K_m$) in the presence of higher concentrations of C10 FA indicate that PlaF undergoes allosteric changes affecting the binding of FAs (*Figure 6b*, *Supplementary file 9*). We examined whether inhibitory FAs affect dimerization by cross-linking of PlaF in the presence of C10, C11, and

C12 FAs. The results of SDS-PAGE revealed a significantly higher amount of dimeric PlaF in FA-treated than in untreated samples (*Figure 6c*). These results suggest a potential regulatory role of FAs on PlaF activity via FA-induced dimerization, which agrees with the previously demonstrated lower activity of the PlaF dimer compared to the monomer (*Figure 5*).

## The tilt of monomeric PlaF in a lipid bilayer permits direct GPL access to the active site

To better understand the molecular mechanism of PlaF activation through monomerization, we performed a set of 10 independent, unbiased 2 µs long MD simulations starting from dimeric or monomeric PlaF embedded in an explicit membrane with a GPL composition similar to the native *P. aeruginosa* membrane (*Figure 7a*). The simulations revealed only minor intramolecular structural changes in monomeric and dimeric PlaF compared to the initial structure (RMSD$_{all atom}$ <4.0 Å) (*Figure 7—figure supplement 1*, *Supplementary file 10*). Spontaneous monomerization was not observed during the MD simulations (*Figure 7—figure supplement 1*), in line with the sub-nanomolar dissociation constant and the simulation timescale. However, in 8 and 6 out of 10 simulations started, respectively, from PlaF$_A$ or PlaF$_B$, a tilting of the monomer for ~25° toward the membrane was observed (*Figure 7b*, left and *Figure 7—figure supplement 1*). This tilting motion cooperatively with rotation of PlaF (*Video 1*) results in the active site cleft of the catalytic domain being oriented perpendicularly to the membrane surface, such that GPL substrates can have direct access to the active site through the opening at the dimer interface (*Figure 7a*, right). In dimeric PlaF, this opening is, according to the model suggested from the X-ray structure, at >5 Å above the membrane surface (*Figure 7a*) so that the diffusion of a GPL from the membrane bilayer to the cleft entrance in this configuration is thermodynamically unfavorable. In all MD simulations started from the tilted PlaF monomer, the protein remains tilted (*Figure 7b*, right and *Figure 7—figure supplement 1*), which corroborates the notion that the tilted orientation is preferred over the respective configuration in di-PlaF.

To further explore the transition of the monomeric PlaF$_A$ to its tilted orientation (t-PlaF$_A$), we calculated the free energy profile or potential of mean force (PMF) for the tilting process by using umbrella sampling and post-processing the distributions with the WHAM method (*Suzuki, 1975*; *Grossfield, 2016*). As reaction coordinate, the distance (*d*) of the top of the JM domain (residues 33–37) to the membrane center was chosen. Distances of ~37 and ~17 Å were calculated for non-tilted PlaF$_A$ using the crystal structure and t-PlaF$_A$ using the structure obtained from the unbiased MD simulations where tilting spontaneously occurred, respectively. The converged and precise (*Figure 7—figure supplement 1*; SEM<0.4 kcal mol$^{-1}$) PMF revealed two minima at *d*=19.6 and 30.6 Å, with t-PlaF$_A$ favored over PlaF$_A$ by 0.66 kcal mol$^{-1}$ (*Figure 7c*). The free energy barrier of ~1.2 kcal mol$^{-1}$ explains the rapid transition from PlaF$_A$ to t-PlaF$_A$ observed in the unbiased MD simulations. The equilibrium constant and free energy of PlaF tilting are $K_{tilting}$=3.35 and a $\Delta G_{tilting}$=–0.8±0.2 kcal mol$^{-1}$. These results suggest a model in which PlaF is activated after monomerization by tilting with respect to the membrane surface, which allows substrate access to its catalytic site.

## Estimating the ratio of monomeric and dimeric PlaF in the cell

To investigate if dimerization-mediated PlaF inhibition is dependent on PlaF concentration in the GPL bilayer, we calculated the free energy profile of dimerization, similarly as for the tilting process. For this, the distance (*r*) between C$_\alpha$ atoms of the JM region of the two chains was used as a reaction coordinate. The converged (*Figure 7—figure supplement 1*) and precise (SEM<1.4 kcal mol$^{-1}$) PMF revealed that di-PlaF is strongly favored at *r*=9.5 Å (–11.4 kcal mol$^{-1}$) over the monomer (*Figure 7d*), fitting with the distance of 9.9 Å observed in the crystal structure of PlaF. From the PMF, the equilibrium constants ($K_a$=1.57×10$^7$ Å$^2$; $K_X$=2.58×10$^5$) and free energy ($\Delta G$=–7.5±0.7 kcal mol$^{-1}$) of PlaF dimerization were computed (*Equations 1–3*), taking into consideration that $K_X$ and $\Delta G$ relate to a state of one PlaF dimer in a membrane of 764 lipids, according to our simulation setup. Experimentally, a concentration of one PlaF dimer per ~3786 lipids in *P. aeruginosa plaF*-overexpressing cells (*Bleffert et al., 2019*) was determined. However, the concentration in *P. aeruginosa* WT is likely 100- to 1000-fold lower, as we could not detect PlaF by Western blot (*Figure 7—figure supplement 2*). Under such physiological conditions and considering that the equilibria for dimer-to-monomer transition and tilting are coupled (*Figure 7a*), between 74% and 96% of the PlaF molecules are predicted

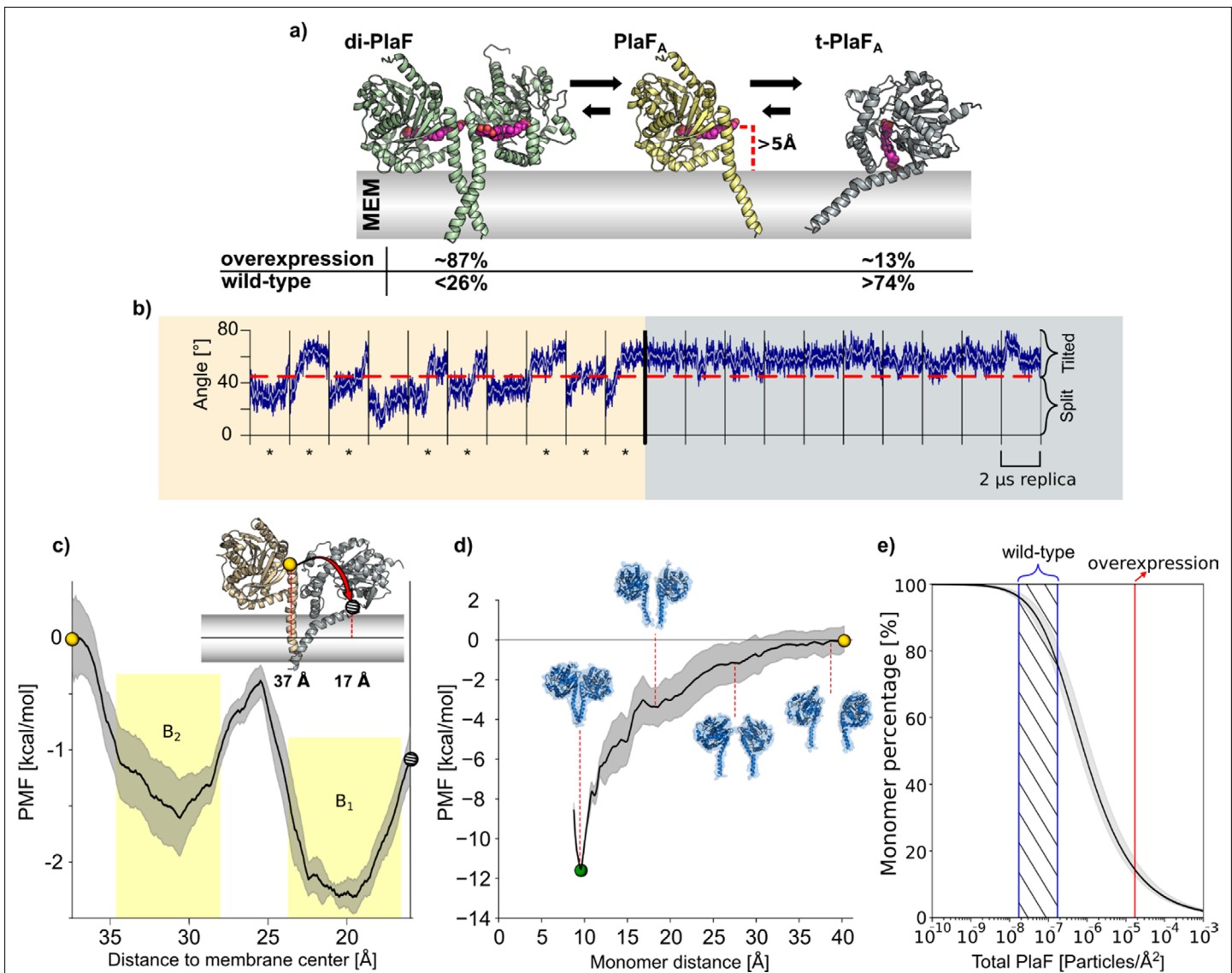

**Figure 7.** Molecular dynamics (MD) simulations and PMF computations of PlaF in the lipid bilayer. (**a**) Structures used for MD simulations. di-PlaF: Crystal structure oriented in the membrane by the PPM method. $PlaF_A$: Chain A from PlaF dimer oriented as in the dimer. The entrance of the active site cleft is more than 5 Å above the membrane bilayer surface. $t$-$PlaF_A$: Extracted monomer A oriented using the PPM method. Cocrystallized MYR, 11A, and OG (depicted in pink), although not included in the simulations, are shown in the figures to highlight the orientation of the active site cleft. Arrows between the structures reflect the predicted states of equilibria under physiological conditions in *Pseudomonas aeruginosa*. Percentages of the different states are obtained from the molecular simulations (see main text and (**e**)). (**b**) MD simulations of monomeric PlaF. Time course of the orientation of monomeric PlaF with respect to the membrane starting from the $PlaF_A$ configuration as observed in the structure (left). In 80% of the trajectories, the monomer ends in a tilted configuration (marked with *). When starting from $t$-$PlaF_A$ (right), in all cases, the structure remains tilted. This shows a significant tendency of the monomer to tilt (McNemar's $X^2$=6.125, p=0.013). (**c**) Potential of mean force (PMF) of monomer tilting. The distance between the COM of $C_\alpha$ atoms of residues 33–37 (yellow, and gray spheres) and the COM of the $C_{18}$ of the oleic acid moieties of all lipids in the membrane (continuous horizontal line in the membrane slab) was used as a reaction coordinate. The shaded area shows the standard error of the mean obtained by dividing the data into four independent parts of 50 ns each. The yellow shaded regions are the integration limits used to calculate $K_{tilting}$ (***Equation 5***). The spheres in the PMF relate to monomer configurations shown in the inset. (**d**) PMF of dimer separation. The distance between the COM of $C_\alpha$ atoms of residues 25–38 of each chain was used as the reaction coordinate. The shaded area shows the standard error of the mean obtained by dividing the data into four independent parts of 50 ns each. Insets show representative structures at intermediate reaction coordinate values. (**e**) Percentage of PlaF monomer as a function of total PlaF concentration in the membrane according to the equilibria shown in (**c**) and (**d**). The monomer percentage was computed according to ***Equations 7–11*** (see Materials and methods and SI for details). The red line shows the experimentally determined PlaF concentration under overexpressing conditions in *P. aeruginosa* p-*plaF*, while the blue-dashed region shows an estimated span for the PlaF concentration in *P. aeruginosa* wild-type (see Materials and methods for details). Calculated percentages are shown in (**a**).

*Figure 7 continued on next page*

**Figure supplement 1.** Structural variations, membrane parameters, and tests for PMF convergence.

**Figure supplement 2.** Detection of PlaF by Western blotting and qRT-PCR.

**Figure supplement 2—source data 2.** Data used to make *Figure 7—figure supplement 2*.

to be in a monomeric, tilted, catalytically active state in *P. aeruginosa* (*Figure 7e*). Our quantitative real-time-PCR results revealed that *plaF* is constitutively expressed in *P. aeruginosa* WT at a much lower level than sigma factors *rpoD* and *rpoS* (*Savli et al., 2003*; *Figure 7—figure supplement 2*). This agrees with previous global proteomic and transcriptomic results (*Erdmann et al., 2018*). As a catalytically active form of PlaF is favored in the WT, PlaF is likely involved in the constant remodeling of membrane GPLs.

## Discussion
### PlaF catalyzed remodeling of membrane GPLs

Employing lipidomic profiling of *P. aeruginosa* WT and the *plaF* gene deletion mutant, we found substantial changes in membrane GPL composition consistent with in vitro PLA$_1$ activity of PlaF and its integral cytoplasmic membrane-localization. The present understanding of bacterial PLAs is limited to extracellular (ExoU, YplA, and SlaA; *Istivan and Coloe, 2006*; *Sawa et al., 2016*) and outer membrane (PlaB and OMPLA; *Snijder et al., 1999*; *Schunder et al., 2010*) enzymes with a proposed role in host-pathogen interactions, but, so far, bacterial PLA proteins tethered to the cytoplasmic membrane were not described (*Jeucken et al., 2019*).

Although bacterial enzymes catalyzing de novo GPL synthesis, their physiological functions and biochemical mechanisms are becoming increasingly well understood (*Jeucken et al., 2019*), information about GPL turnover enzymes remains largely obscure. Several of our findings indicate that PlaF plays a hitherto unexplored role in the membrane remodeling (*Figure 8*) that becomes especially apparent during virulence adaptation.

i. Deletion of *plaF* gene in *P. aeruginosa* leads to accumulation of several low abundant PE, PG, and PC molecular species (*Figure 2c*). PE, PG, and PC with different acyl chain lengths (C12–C18) were hydrolyzed by PlaF in vitro (*Figure 2b*). A low in vitro PLA$_1$ activity of PlaF (µU/mg) is expected for an enzyme that could irreversibly damage the membrane.

ii. The *P. aeruginosa* Δ*plaF* strain revealed several depleted GPLs (*Figure 2c*), which may be explained assuming that lysoGPLs generated by PlaF activity are missing in this strain for further acylation to yield modified GPLs.

iii. FAs with 10–14 carbon atoms inhibit PlaF activity in vitro (*Figure 6a*). As PlaF can produce such FAs in vivo (*Figure 2c*), it is reasonable to assume that their cellular function is related to the regulation of PlaF activity by product feedback inhibition. This phenomenon is well known for lipolytic (*Ruiz et al., 2004*; *Markweg-Hanke et al., 1995*) and other central metabolic enzymes (*Rose, 1971*; *Van Schaftingen and Hers, 1981*; *Alam et al., 2017*).

iv. PlaF is constitutively expressed (*Figure 7—figure supplement 2* and *Erdmann et al., 2018*) at low levels suggesting that PlaF-catalyzed GPL remodeling may have general importance for *P. aeruginosa* physiology.

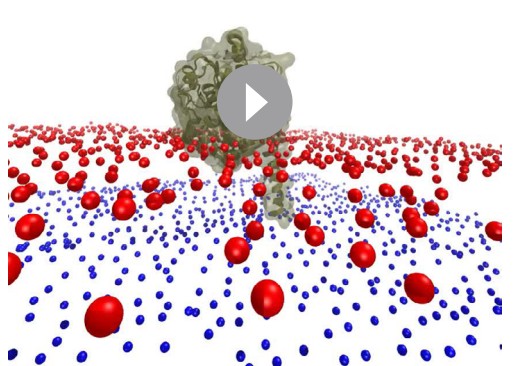

**Video 1.** MD simulation of monomeric PlaFA in GPL bilayer. Blue and red spheres indicate head groups of GPLs in two leaflets of the bilayer.

https://elifesciences.org/articles/72824/figures#video1

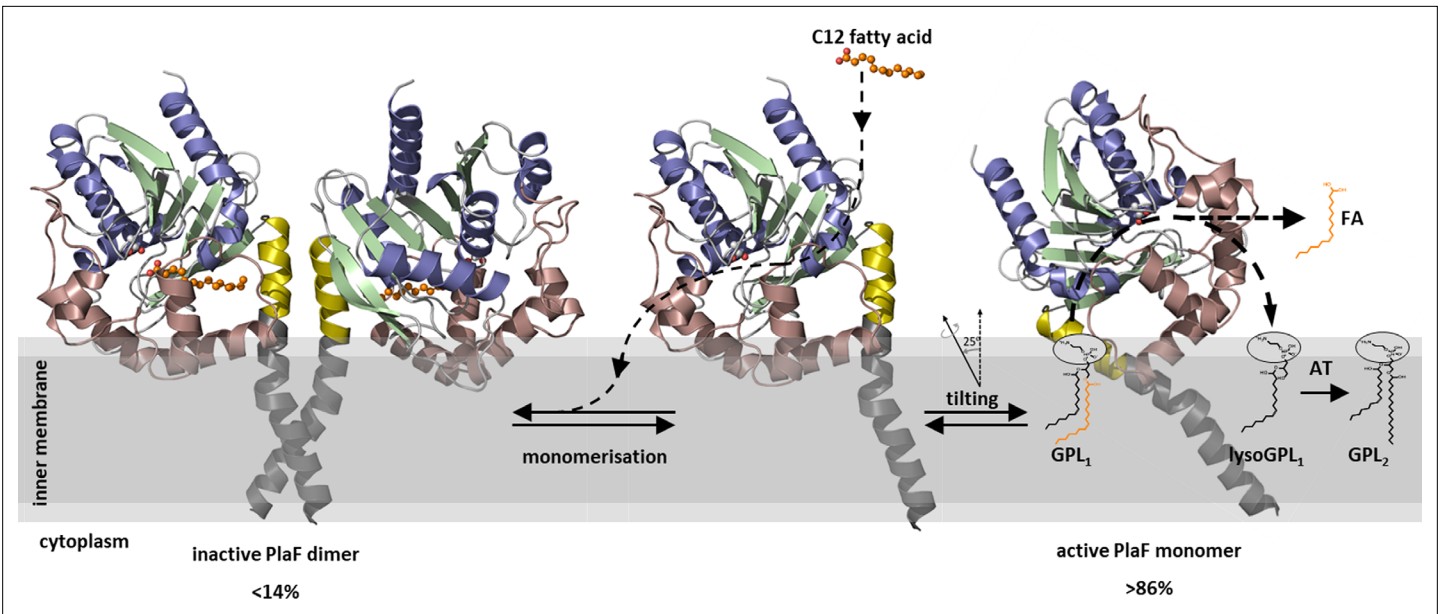

**Figure 8.** A model of PlaF-mediated membrane GPL remodeling. PlaF is anchored with the TM helix to the inner membrane of *Pseudomonas aeruginosa* (*Figures 1c and 4c*), where it forms an inactive dimer (*Figure 5c*). Monomerization (*Figure 5c*) and subsequent spontaneous tilting (*Figure 7*) lead to activation. Binding of dodecanoic acid (C12) to monomeric PlaF triggers dimerization (*Figure 6c*) and inhibits enzymatic activity (*Figure 6a*). Tilting constrains the active site cavity of PlaF to the membrane surface such that GPL substrates can enter (GPL$_1$, *Figure 2*), which are hydrolyzed to FA and lysoGPL$_1$. A yet unknown acyl transferase possibly acylates lyso-GPL$_1$ to yield modified GPL$_2$ (*Figure 2*). GPL, glycerophospholipid.

v. The *P. aeruginosa* Δ*plaF* strain shows strongly impaired killing of *G. mellonella* and human macrophages compared to WT (*Figure 3*), thus revealing the important function of PlaF-mediated GPL remodeling for *P. aeruginosa* virulence.

It is well known that the global diversity of GPL acyl chains in eukaryotes derives from de novo synthesis (Kennedy pathway) and remodeling (Lands cycle) pathways, which are differentially regulated (*Jacquemyn et al., 2017*). In the Lands cycle, GPLs are targeted by PLA and acyltransferases that respectively remove and replace acyl chains in GPLs by a recently described mechanism (*Zhang et al., 2021*; *Mouchlis et al., 2015*). We suggest that PlaF is the PLA that alters *P. aeruginosa* membranes by hydrolysis of the main classes of GPLs, namely PE, PG, and PC. Although the observed changes may be caused by the absence of PlaF in the membrane of *P. aeruginosa*, it is more likely that PlaF directly hydrolyses GPLs as only low concentrations of PlaF were detected in the cell (*Figure 7—figure supplement 2*). The exact molecular function of PlaF in GPL-remodeling and the regulation of virulence of *P. aeruginosa* remains unknown. One possibility is that PlaF tunes the concentration of low-abundance GPL species in the membrane, creating a suitable membrane environment for the stabilization of membrane proteins or protein complexes (*Corradi et al., 2019*). In addition, PlaF-generated GPLs might have a more sophisticated function for membrane-embedded virulence-related proteins. This was demonstrated for ABHD6, a human membrane-bound PLA, which controls the membrane concentration of lipid transmitter 2-arachidonoylglycerol involved in regulating the endocannabinoid receptor (*Marrs et al., 2010*). Notably, human ABHD6 and PlaF share ~50% sequence similarity and hydrolize similar substrates (*Bleffert et al., 2019*).

Although PlaF is an important enzyme involved in GPL metabolism, future research should reveal (i) which acyltransferase is involved in the acylation of lysoGPLs produced by PlaF, (ii) if PlaF has acyltransferase activity as described for cPLA$_2$γ involved in the Lands cycle in humans (*Asai et al., 2003*), and (iii) if periplasmic lysophospholipase TesA (*Kovačić et al., 2013*) and the recently discovered intracellular PLA PlaB (*Weiler et al., 2022*) are involved in the Lands cycle.

## Structural insights into dimerization and ligand-mediated regulation of PlaF activity

The high-resolution structure of PlaF with the natural ligands (FAs) bound to its active site represents the first dimeric structure of a full-length, single-pass TM protein (*Figure 4*). It contributes to our understanding of the role of TM-JM domain-mediated dimerization for the biological activity of single-pass TM proteins, which is undisputed in bacteria and eukaryotes, yet, poorly understood at the atomic level due to the lack of full-length dimeric structures (*Bocharov et al., 2017*; *Fink et al., 2012*). The present structure-function relationship of single-pass TM dimers derives from structural data of isolated TM helices without their soluble domains. Therefore, their biological relevance remains questionable (*Bocharov et al., 2017*).

The crystal structure of PlaF reveals unprecedented details of interactions between the membrane-spanning TM-JM domains and underlines the role of PlaF for degradation of membrane GPLs. The TM and JM domains are not distinct but fold into a long kinked α-helix (*Figure 4a*). This is different from the structure of a human epidermal growth factor receptor (EGFR), the only structure of an isolated TM-JM domain, in which TM and JM helices are connected by an unstructured loop (*Bragin et al., 2016*; *Endres et al., 2013*). The mechanism undergoing PlaF dimerization likely differs from the EGFR family, although it is not excluded that the truncation of soluble domains might destabilize the TM-JM dimer of EGFR, leading to structural changes. We identified intramolecular interactions of 13 residues from the catalytic domain of PlaF with the JM domain, which clearly demonstrates the stabilizing role of the soluble domain on the TM-JM helix. Sole interactions of TM-JM helices result in the formation of a coiled-coil structure (*Figure 4b*) that stabilizes the PlaF dimer by burying the surface of 656 Å$^2$, which is slightly larger than the interface of the glycophorin TM helix dimer (400 Å$^2$) without the JM region (*MacKenzie et al., 1997*). The biological relevance of PlaF dimerization is corroborated by crosslinking experiments with *P. aeruginosa* cells, which revealed the in vivo occurrence of PlaF dimer (*Figure 5a*). Furthermore, enzyme activity measurements and MST analysis of protein-protein interactions revealed that the activity decreases and dimerization increases as a function of increasing PlaF concentration in vitro (*Figure 5c*). These findings open the question of regulation of dimerization-mediated PlaF inhibition in vivo and the role of membrane GPLs and their hydrolytic products in this process. Homodimerization mediated via TM-JM interactions was previously shown to be required for activation of single-span TM proteins from receptor tyrosine kinase (*Li and Hristova, 2010*) and ToxR-like transcriptional regulator (*Buchner et al., 2015*). However, structural and mechanistic details remained unknown.

A metabolic role of PlaF related to the liberation of FAs and lysoGPLs from membrane GPL substrates addresses the question of regulating PlaF function by substrates or products. A dimer interface with mainly hydrophobic interactions and a few H-bonds detected in the JM region (*Figure 4b*) seems to be designed to interact with amphipathic GPLs. However, it remains to be elucidated if PlaF-GPL interactions regulate PlaF dimerization and its activity as shown for interactions of SecYEG with cardiolipin and bacteriorhodopsin with sulfated tetraglycosyldiphytanylglycerol (*Corradi et al., 2019*; *Essen et al., 1998*).

C10–C14 FAs exert competitive inhibition as in vitro effectors of PlaF (*Figure 6a*) and enhance dimerization (*Figure 6c*) in the concentration range (0.5–7.5 mM) similar to the intracellular concentration of FAs in *E. coli* (~2–4 mM) (*Lennen et al., 2013*). The dimerization-triggering function of FAs is strengthened by observing a mixed-type inhibition (*Figure 6b*), which indicates that FAs affect PlaF not only by binding to the active site but also by modulating the oligomerization equilibrium (*Gabizon and Friedler, 2014*). Interestingly, we identified FA ligands in the PlaF structure bound to the PlaF active site cleft (*Figure 4*) that were copurified with PlaF from *P. aeruginosa* (*Figure 4—figure supplement 1*). Furthermore, we identified an OG molecule, used for purification, in the active site of PlaF. The pseudo-ligand (OG) and natural products (FAs) form an intricate interaction network connecting the catalytic ($S_{137}$) with the dimerization site ($S_{29}$, $T_{32}$, and $V_{33}$) in the JM domain (*Figure 4e*). Although the static structure of dimeric PlaF cannot explain how FAs trigger dimerization, we speculate that in vivo, the position of the OG molecule is occupied by FAs, which facilitates the interaction between the two JM-helices, stabilizing the dimer.

## Atomistic model of PlaF catalyzed hydrolysis of membrane GPLs

The question remains of how does the PlaF dimer-to-monomer transition activate PlaF in the GPL bilayer? The active sites in the crystal structure of di-PlaF already adopt catalytically active conformations (*Figure 4a*), suggesting that the activation of PlaF most likely does not involve structural rearrangements of the active site. To unravel a possible effect of the structural dynamics of PlaF in the membrane on enzyme regulation by dimerization, we performed extensive MD simulations and configurational free energy computations on dimeric and monomeric PlaF embedded into a GPL bilayer mimicking the bacterial cytoplasmic membrane. While structural changes within di-PlaF and monomeric PlaF were moderate (*Supplementary file 10*), monomeric PlaF spontaneously tilted as a whole toward the membrane, constraining the enzyme protein in a configuration with the opening of the active site cleft immersed into the GPL bilayer (*Figure 7a and b*). A configuration similar to t-PlaF was observed for monomeric *Saccharomyces cerevisiae* lanosterol 14α-demethylase, a single TM spanning protein acting on a membrane-bound substrate (*Monk et al., 2014*). In t-PlaF, GPL can access the active site cleft directly from the membrane with the sn-1 acyl chain entering the first (*Wittgens et al., 2017*). This is unlikely in di-PlaF, in which the opening of the active site cleft is >5 Å above the membrane (*Figure 7e*). There, a GPL would need to leave the bilayer into the water before entering the active site cleft, which is thermodynamically unfavorable.

Based on the experimental evidence, we propose a hitherto undescribed mechanism by which the transition of PlaF between a dimeric, not-tilted to a monomeric, tilted configuration is intimately linked to the modulation of the PlaF activity. This mechanism, to the best of our knowledge, expands the general understanding of mechanisms of inactivation of integral single-pass TM proteins and differs from suggested allosteric mechanisms implying structural rearrangements (even folding), mostly in the JM domain, upon ligand binding as underlying causes for functional regulation (*Bocharov et al., 2017*). Rather, for PlaF, monomerization followed by a global reorientation of the single-pass TM protein in the membrane is the central, function-determining element.

Based on computed free energies of association (*Figure 7d*) and tilting (*Figure 7c*), and taking into account the concentration range of PlaF in *P. aeruginosa*, PlaF preferentially exists as t-PlaF in the cytoplasmic membrane (*Figure 7e*). Increasing the PlaF concentration in the membrane will thus shift the equilibrium toward di-PlaF. This result can explain the observations that PlaF, an enzyme with membrane-disruptive activity, is found in only very low amounts (*Figure 7—figure supplement 2*) in WT *P. aeruginosa* cells and that overproduction of PlaF in *P. aeruginosa* is not harmful to the cells.

## Implications for drug development

Based on our observation that *P. aeruginosa* Δ*palF* shows strongly attenuated virulence, we suggest that interfering with PlaF function might be a promising target for developing new antibiotics against *P. aeruginosa*. This class of antibiotics should be potent assuming that GPL remodeling plays a global role in the virulence adaptation in bacteria through simultaneous regulation of virulence-related processes (*Benamara et al., 2014*; *Le Sénéchal et al., 2019*; *El Khoury et al., 2017*; *Blanka et al., 2015*). Analogously, eukaryotic PLAs regulating inflammatory pathways through the release of arachidonic acid were recently suggested as potential targets of anti-inflammatory drugs (*Mouchlis and Dennis, 2016*). Our structural and mechanistic studies provide a basis for targeting PlaF by competitive inhibition and interfering with dimerization (*Gabizon and Friedler, 2014*; *Hopkins and Groom, 2002*).

## Materials and methods

### Cloning, protein expression, and purification

Molecular biology methods, DNA purification, and analysis by electrophoresis were performed as described previously (*Kovacic et al., 2016*). For the expression of PlaF, *P. aeruginosa* PAO1 (WT) cells transformed (*Choi et al., 2006*) with plasmid pBBR-*pa2949* (*Kovacic et al., 2016*), here abbreviated as p-*plaF*, were grown overnight at 37°C in lysogeny broth (LB) medium supplemented with tetracycline (100 µg/ml) (*Bleffert et al., 2019*). The total membrane fraction of *P. aeruginosa* p-*plaF* was obtained by ultracentrifugation, membranes were solubilized with Triton X-100, and PlaF was purified using Ni-NTA IMAC and buffers supplemented with 30 mM OG (*Bleffert et al., 2019*). For biochemical analysis, PlaF was transferred to Tris-HCl (100 mM, pH 8) supplemented with 30 mM OG (*Table 2*).

**Table 2.** Material used in this work.

| Material | Ordering details |
|---|---|
| *Galleria mellonella* larvae | Fauna Topics GmbH, order number: 527 |
| Trypsin, porcine, MS grade | Merck, order number: 650279 |
| Anti-SecG antiserum | gift of R. Voulhoux, CNRS AMU LCB, Marseille |
| Anti-lipid A antibodies | Acris Antibodies, Herford, Germany, order number: BP 2235 |
| Ni-NTA agarose | Macherey–Nagel, Düren, Germany, order number: 745400 |
| *n*-Octyl-β-D-glucoside | Merck, order number: 850511P |
| *para*-Nitrophenyl butyrate | Sigma-Aldrich, order number: N9876 |
| Glycerophospholipids | Avanti Polar Lipids, Alabaster, USA |
| NEFA-HR(2) kit | Wako Chemicals, Richmond, USA, order number: 999-34691 |
| [N-((6-(2,4-DNP)amino)hexanoyl)-1-(BODIPYFL C5)-2-hexyl-sn-glycero-3-phosphoethanolamine] | Thermo Fisher Scientific Inc, Waltham, MA, order number: A10070 |
| 1-O-(6-BODIPY558/568-aminohexyl)-2-BODIPYFL C5-Sn-glycero-3-phosphocholine | Thermo Fisher Scientific Inc, Waltham, MA, order number: A10072 |
| N-((6-(2,4-dinitrophenyl)amino)hexanoyl)-2-(4,4-difluoro-5,7-dimethyl-4-bora-3a,4a-diaza-s-indacene-3-pentanoyl)–1-hexadecanoyl-sn-glycero-3-phosphoethanolamine triethylammonium salt | Thermo Fisher Scientific Inc, Waltham, MA, order number: D23739 |
| *Thermomyces lanuginosus* PLA$_1$ | Sigma-Aldrich, order number: L3295 |
| *Naja mocambique mocambique* PLA$_2$ | Sigma-Aldrich, order number: P7778 |
| Dimethyl pimelimidate | Merck, order number: 80490 |
| Bis(sulfosuccinimidyl) glutarate | Thermo Fisher Scientific, order number: 21610 |
| Bis(sulfosuccinimidyl) suberate | Thermo Fisher Scientific, order number: 21586 |
| PD-10 columns | Merck, order number: GE17-0851-01 |
| NHS Labeling Kit | NanoTemper, Munich, Germany, order number: MO-L011 |
| 4-Methylumbelliferyl palmitate | Sigma-Aldrich; order number: M7259 |
| CytoTox 96 non-radioactive cytotoxicity assay | Promega, order number: G1780 |
| NucleoSpin RNA Preparation Kit | Macherey–Nagel, Düren, Germany, order number: 740955 |
| RNase-Free DNase Kit | Qiagen, Hilden, Germany, order number: 79254 |
| Ambion DNA-Free DNase Kit | Thermo Fisher Scientific, Darmstadt, Germany, order number: AM1906 |
| Maxima First Strand cDNA Synthesis Kit | Thermo Fisher Scientific, Darmstadt, Germany, Order number: K1641 |
| SYBR Green/ROX qPCR Master Mix | Thermo Fisher Scientific, Darmstadt, Germany, order number: K0221 |
| N-Methyl-N-(trimethylsilyl) trifluoroacetamide | Sigma-Aldrich; order number: 69479 |

## SDS-PAGE, zymography, and immunodetection

The protein analysis by electrophoresis under denaturation conditions (*Laemmli, 1970*), in-gel esterase activity (zymography), and immunodetection by Western blotting were performed as described previously (*Kovacic et al., 2016*). The protein concentration was determined by UV spectrometry using a theoretical extinction coefficient of PlaF containing a C-terminal His$_6$-tag of 22,920 M$^{-1}$ cm$^{-1}$ (*Bleffert et al., 2019*).

## Enzyme activity assays, inhibition, and enzyme kinetic studies

Esterase activity assays with *p*-nitrophenyl FA esters as substrates were performed in 96-well microtiter plates as described previously (*Kovacic et al., 2016*). Phospholipid substrates purchased from Avanti Polar Lipids (*Table 2*) were prepared for PLA activity assays (25 µl enzyme+25 µl substrate) performed as described previously (*Jaeger and Kovacic, 2014*). The amount of FAs released by the PLA activity of PlaF was determined using the NEFA-HR(2) Kit (Wako Chemicals, Richmond, USA) (*Bleffert et al., 2019*). PLA$_1$ and PLA$_2$ activities of PlaF were measured using fluorescent substrates purchased from Thermo Fisher Scientific Inc (*Table 2*): PLA1-PE, [N-((6-(2,4-DNP)amino)hexanoyl)-1-(BODIPYFL C5)-2-hexyl-sn-glycero-3-phosphoethanolamine]; PLA2-PC, 1-O-(6-BODIPY558/568-aminohexyl)-2-BODIPYFL C5-Sn-glycero-3-phosphocholine; and PLA2-PE, *N*-((6-(2,4-dinitrophenyl)amino)hexanoyl)-2-(4,4-difluoro-5,7-dimethyl-4-bora-3a,4a-diaza-*s*-indacene-3-pentanoyl)-1-hexadecanoyl-*sn*-glycero-3-phosphoethanolamine triethylammonium as described by *da Mata Madeira et al., 2016*. Measurements were performed using a plate reader in 96-well plates at 25°C by combining 50 µl of the substrate with 50 µl PlaF (0.7 µg/ml), or control enzymes, the PLA$_1$ of *Thermomyces lanuginosus* (5 U/ml) and the PLA$_2$ or *Naja mocambique mocambique* (0.7 U/ml).

## Inhibition

The inhibition of PlaF by FAs was assayed by combining FA dissolved in DMSO (20-fold stock solution) with *para-n*itrophenyl butyrate (*p*-NPB)substrate solution followed by the addition of the PlaF sample (8 nmol) and spectrophotometric enzyme activity measurement using *p*-NPB substrate (*Tian and Tsou, 1982*). In control experiments, all compounds except FA were combined to assess PlaF activity in the absence of FA. Inhibition constants were calculated by fitting enzyme kinetic parameters obtained by varying FA concentration (0, 0.5, 1.5, 2.5, 5, and 7.5 mM) for different substrate concentrations (0.05, 0.1, 0.2, 0.3, 0.5, and 1 mM) (*Kenakin, 2012*).

## Subcellular localization

Membranes from *P. aeruginosa* WT and p-*plaF* (PlaF overproduction strain) were isolated as described previously (*Kovacic et al., 2016*). To separate integral from peripheral membrane proteins, total cell membranes were incubated for 30 min at room temperature with: 10 mM Na$_2$CO$_3$ (pH 11), 4 M urea (in 20 mM MES buffer pH 6.5) or 2% (w/v) Triton X-100 (in 20 mM MES buffer pH 6.5). After the incubation, the samples were centrifuged for 30 min at 180,000$g$ to separate membranes from solubilized proteins.

The separation of the inner and outer membrane was performed with a discontinuous sucrose gradient by ultracentrifugation at 180,000$g$ for 72 hr and 4°C (*Viarre et al., 2009*). The sucrose gradient consisted of 1.5 ml fractions with 35%, 42%, 46%, 50%, 54%, 58%, 62%, and 65% (w/v) sucrose in 100 mM Tris-HCl, pH 7.4. Isolated membranes from *P. aeruginosa* WT were suspended in buffer containing 35% (w/v) sucrose and loaded on the top of the discontinuous sucrose gradient. Fractions were collected from the bottom (pierced tube), and sucrose concentration was determined with a refractometer (OPTEC, Optimal Technology, Baldock, UK). To determine the orientation of catalytic PlaF domain *P. aeruginosa* p-*plaF* cells (10 ml culture with OD$_{580nm}$ 1 grown in LB medium at 37°C) were harvested by centrifugation (4000$g$, 4°C, 5 min) and suspended in 1 ml Tris-HCl buffer (50 mM, pH 7.5, 10% sucrose (w/v)) followed by shock freezing with liquid nitrogen (*Eichler and Wickner, 1998*). Cells were thawed to room temperature and centrifuged (4000$g$, 4°C, 5 min) followed by incubation of the pellet for 1 hr on ice in Tris-HCl buffer (30 mM, pH 8.1, sucrose 20% (w/v) EDTA 10 mM). Trypsin (20 µl, 1 mg/ml) was added to the suspension containing the cells with the permeabilized outer membrane and incubated at room temperature for up to 5 hr. The proteolytic reaction was stopped with onefold SDS-PAGE sample buffer and incubation for 10 min at 99°C. Immunodetection of SecG with anti-SecG antiserum (gift of R. Voulhoux, CNRS AMU LCB, Marseille) and lipid A antibodies (BP 2235, Acris Antibodies, Herford, Germany) was performed as described above for PlaF using the respective antisera at 1/2000 and 1/1000 dilutions.

## Cross-linking assays

In vitro cross-linking using the bifunctional cross-linking reagents DMP was performed as previously described (*de Jong et al., 2017*) with the following modifications. PlaF (10 µl, 15.5 µM) purified with

OG was incubated with 6 µl freshly prepared DMP (150 mM in 100 mM Tris-HCl, pH 8.4), $BS^2G$ (5 mM in 100 mM Tris-HCl, pH 8.0) and $BS^3$ (5 mM in 100 mM Tris-HCl, pH 8.0) for 90 min (*Table 2*). The cross-linking reaction was terminated with a 5 µl stop solution (50 mM Tris-HCl, 1 M glycine, NaCl 150 mM, pH 8.3). For in vivo cross-linking, *P. aeruginosa* p-*plaF* and EV strains were grown in LB medium at 37°C to $OD_{580nm}$ 1. Cells were harvested by centrifugation (10 min, 4000*g*, 4°C), suspended in 1/20 volume of Tris-HCl (pH 8.3, 100 mM, NaCl 150 mM), and treated with the same volume of freshly prepared cell-permeable cross-linking reagent DMP (0, 20, 30, and 50 mM in Tris-HCl buffer 100 mM, pH 8.4) for 2 h. The cross-linking reaction was terminated with the same volume of stop solution (50 mM Tris-HCl, 1 M glycine, 150 mM NaCl, pH 8.3).

## Analysis of concentration-dependent dimerization

Purified PlaF (20 µl, 50–60 µM) was transferred from the purification buffer into the labeling buffer (Na-$PO_4$ 20 mM, pH 8.3) supplemented with OG (30 mM) using PD-10 columns (GE Healthcare, Solingen, Germany) according to the manufacturer's protocol. Labeling was performed by incubating PlaF with 15 µl dye (440 µM stock solution) for 2.5 hr using the NHS Labeling Kit (*Table 2*). PlaF was then transferred into a purification buffer using PD-10 columns. Non-labeled PlaF was diluted with the same buffer in 16 steps by combining the same volume of the protein and buffer, yielding samples with concentrations from 26.9 µM to 1.6 nM. Samples containing 100 nM labeled PlaF were incubated for 16 hr at room temperature in the dark, and MST experiments were performed using the Monolith NT.115 device (NanoTemper, Munich, Germany) with the following setup: MST power, 60%; excitation power 20%; excitation type, red; 25°C. Constants were calculated according to the four-parameter logistic, nonlinear regression model using Origin Pro 2018 software.

The enzymatic activity of PlaF samples used for MST analysis was assayed by combing 15 µl of enzyme and 15 µl 4-methylumbelliferyl palmitate (4-MUP, 2 mM) dissolved in purification Tris-HCl (100 mM, pH 8) containing 10% (v/v) propan-2-ol (*Table 2*). Fluorescence was measured for 10 min (5 s steps) using a plate reader in black 96-well microtiter plates at 30°C.

## Construction of a *P. aeruginosa* Δ*plaF*, and Δ*plaF::plaF* strains

The mutagenesis vector pEMG-Δ*plaF* (*Figure 2—figure supplement 2*) was generated with upstream and downstream regions of *plaF* gene amplified by standard PCR using Phusion DNA polymerase, a genomic DNA of *P. aeruginosa* PAO1 as a template, and primer pairs 5'-ATATATGAATTCTCTGCTCG GCGCGAAACGCAGCGP-3'/5'-ATATATACGCGTGGGTGTCCGAAGGCTTCAGGAAAAAAGGGGC-3' and 5'-ATATATACGCGTAAACGCGAACCGGCGCCTGGG-3'/5'-CTGGATGAATTCTGGCCTGGACAC CGACAAGGAAGTGATCAAGG-3', respectively. DNA fragments upstream and downstream of the *plaF* gene were cloned into the pEMG vector by ligation of DNA fragments hydrolyzed with *Eco*RI restriction endonuclease. *P. aeruginosa* PAO1 (WT) cells were transformed with the pEMG-Δ*plaF* and *P. aeruginosa* Δ*plaF* mutant strain was generated by homologous recombination (*Martínez-García and de Lorenzo, 2011*). Generation of pUC18T-mini-Tn7T-Gm-*plaF* plasmid (*Figure 2—figure supplement 2*) for recombination of *plaF* gene containing 128 bp upstream region of *plaF* with a chromosome of *P. aeruginosa* Δ*plaF*. A DNA fragment containing the upstream region and *plaF* gene was amplified using primer pair 5'-AATAGAGCTCACCGCCGTCCTTAGGTTC-3'/5'-AATAGAGCTCCGTTTTCAGC GACCGGC-3' from the genomic DNA of *P. aeruginosa* PAO1. Both primers contained the restriction site *Sac*I for cloning into the pUC18T-mini-Tn7T-Gm (gifts from Herbert Schweizer, Addgene plasmids #63121, #64968, and #64946). *P. aeruginosa* Δ*plaF* was transformed with pUC18T-mini-Tn7T-*plaF*-Gm and helper plasmid pTNS2 encoding the Tn7 site-specific transposase ABCD by tri-parental conjugation and the positive clones were identified by PCR using primer pair 5'-GCACATCGGCGACGTG CTCTC-3'/5'-CACAGCATAACTGGACTGATTTC-3'. The gentamycin-resistance gene was excised from *P. aeruginosa* Δ*plaF::plaF*-Gm by Flp-recombinase produced from pFLP3 plasmid (*Choi et al., 2005*).

## *G. mellonella* virulence model

*G. mellonella* larvae (*Table 2*) were sorted according to size and split into groups of 10 in Petri dishes. *P. aeruginosa* WT, the Δ*plaF*, and the Δ*plaF::plaF* strains were grown overnight and sub-cultured to mid-log phase in LB media at 37°C. The bacteria were washed twice with PBS and adjusted to $OD_{600}$ 0.055, which equals $5 \times 10^4$ bacteria/µl. This suspension was diluted in PBS to the infection dose of 500 bacteria per 10 µl, which were injected into the hindmost left proleg of the insect. Hereby, PBS

injections were used as infection control and untreated larvae as viability control. If more than one larvae was dying within the control group, the experiment was repeated. The survival of larvae incubated at 30°C was monitored (*Koch, 2014*).

## Cytotoxicity assay

BMDMs were isolated from the bones of C57BL/6 mice and cultured in RPMI supplemented with 20% (v/v) conditioned L929 medium to allow for differentiation into macrophages for at least 7 days. BMDMs were seeded at a concentration of $5 \times 10^5$ cells in a 24-well plate. The BMDMs cells (n=10) were infected with $5 \times 10^5$ bacteria (cultivated overnight in LB medium at 37°C), which accounts for MOI 1 (*Mittal et al., 2016*). PBS treated cells served as viability control. Supernatants were taken at 0, 1, 3, and 6 hr post-infection. LDH levels were determined (n=6) using the CytoTox 96 Non-Radioactive Cytotoxicity Assay according to the manufacturer's protocol. As 100% killing control, uninfected cells were lysed with 1% (v/v) Triton-X100. Statistical analysis was performed using a one-way ANOVA to determine significant changes of normally distributed values obtained from two independent experiments with 10 samples each.

## Growth curves

The growth of *P. aeruginosa* WT and Δ*plaF* cultures in Erlenmeyer flasks (agitation at 160 rpm) was monitored by measuring $OD_{580nm}$ for 24 hr. $OD_{580nm}$ was converted to colony-forming units by multiplying with the factor $8 \times 10^8$ experimentally determined for *P. aeruginosa* PAO1 strain from our laboratory.

## Quantitative real-time-PCR

RNA was isolated from *P. aeruginosa* PA01 and Δ*plaF* grown overnight (37°C, LB medium) with the NucleoSpin RNA Preparation Kit and genomic DNA was quantitatively removed using RNase-Free DNase Kit and Ambion DNA-Free DNase Kit according to the manufacturer's recommendations (*Table 2*). One µg of RNA was transcribed into cDNA using the Maxima First Strand cDNA Synthesis Kit (*Table 2*). For the quantitative real-time-PCR (qRT-PCR), 50 ng of cDNA was mixed with SYBR Green/ROX qPCR Master Mix (*Table 2*) to a total volume of 20 µl and qRT-PCR was performed as described previously (*Savli et al., 2003*). Following primers were used for *rpoD* (3'-CAGCTCGA CAAGGCCAAGAA-5', CCAGCTTGATCGGCATGAAC), *rpoS* (3'-CTCCCCGGGCAACTCCAAAAG-5', 3'-CGATCATCCGCTTCCGACCAG-5') and *plaF* (3'-CGACCCTGTTGCTGATCCAC-5', 3'-ACGTCGTA GCTGGCCTGTTG-5').

## Lipidomic analysis of GPLs extracted from cell membranes

The cells of *P. aeruginosa* WT, Δ*plaF*, and Δ*plaF::plaF* cultures grown overnight in 15 ml LB medium (*Supplementary file 3*) at 37°C were harvested by centrifugation at 4000*g* and 4°C for 15 min and suspended in 2 ml ddH$_2$O followed by boiling for 10 min to inactivate phospholipases. Cells were harvested by centrifugation (4000*g*, 4°C, 15 min) and total lipids were extracted from the cell pellet (*Gasulla et al., 2013*). Briefly, after boiling the water was removed by centrifugation (4000*g*, 4°C, 15 min). Lipids were extracted with CHCl$_3$:CH$_3$OH=1:2 (v/v) and the organic phase was collected. The extraction was repeated with CHCl$_3$:CH$_3$OH=2:1 (v/v) and the organic phases were combined. One volume of CHCl$_3$ and 0.75 volumes of an aqueous solution containing 1 M KCl and 0.2 M H$_3$PO$_4$ were added to the combined chloroform/methanol extracts. Samples were vortexed and centrifuged (2000*g*, 5 min). The organic phase was withdrawn and the solvent of the lipid extract was evaporated under a stream of N$_2$. Total lipids were dissolved in CHCl$_3$:CH$_3$OH=2:1 (v/v). GPLs were quantified by Q-TOF mass spectrometry (Q-TOF 6530; Agilent Technologies, Böblingen, Germany) as described elsewhere (*Gasulla et al., 2013*). Statistical analysis of the GPL amount was performed using the T-test and the Shapiro-Wilk method to determine significant changes of normally distributed values obtained from four *P. aeruginosa* WT lipidome and four Δ*plaF* samples. Ratio of PlaF and GPLs was calculated knowing GPLs extraction yield of 40 µg GPLs per 1 ml *P. aeruginosa* p-*plaF* ($OD_{580nm}$ 1) and PlaF purification yield of ~1 µg from 1 ml *P. aeruginosa* p-*plaF* culture with $OD_{580nm}$ 1 (*Bleffert et al., 2019*).

## GC-MS analysis of FA

FAs were extracted from PlaF purified from 13 g *P. aeruginosa* p-*plaF* cells with OG using four parts of organic solvent (CHCl$_3$:CH$_3$OH=2:1). Extraction was repeated three times, the chloroform extracts

were combined, chloroform was evaporated, and FAs were dissolved in 200 µl chloroform. The chloroform extract was mixed with 10 volumes of acetonitrile and filtered through a 0.2 µm pore size filter. For GC-MS analysis, FA extracts and standards (C10-, C11-, C14-, C15-, C16-, and C18-FA; C16-, C18-, and C20-primary fatty alcohol) were converted into their trimethylsilyl esters and trimethylsilyl ethers, respectively. 900 µl of the sample or standard solution (CHCl$_3$:acetonitrile=1:5) was mixed with 100 µl N-methyl-N-(trimethylsilyl) trifluoroacetamide and heated to 80°C for 1 hr. The GC-MS system consisted of a Trace GC Ultra gas chromatograph, TriPlus autosampler, and an ITQ 900 mass spectrometer (Thermo Fisher Scientific, Waltham, MA). Analytes were separated on a Zebron-5-HT Inferno column (60 m × 0.25 mm i.d., 0.25 µm film thickness, Phenomenex, USA). Helium was used as carrier gas at a constant gas flow of 1.0 ml/min. The oven temperature program was as follows: 80°C; 5°C/min to 340°C, held for 5 min. The injector temperature was held at 290°C, and all injections (1 µl) were made in the split mode (1:10). The mass spectrometer was used in the electron impact (EI, 70 eV) mode and scanned over the range m/z 25–450 with an acquisition rate of 3 microscans. The transfer line and ion source were both kept at 280°C. Data processing was performed by the use of the software XCalibur 2.0.7 (Thermo Fisher Scientific). FAs from the PlaF sample were identified by comparison of their retention times and mass spectra with FA standards.

Reaction of purified PlaF (620 µl, 300 µg/ml) with 1-(9Z-octadecenoyl)-2-pentadecanoyl-glycero-3-phospho-(1'-rac-glycerol) (PG$_{15:0-18:1}$, 0.5 mM) in 4 ml NEFA buffer was conducted for 24 hr at 37°C followed by extraction of FAs, derivatization, and GC quantification. FAs were transferred to 15 ml Falcon tubes by dissolving in 500 µL CH$_2$Cl$_2$ twice. After evaporation to dryness the remaining fatty acids were derivatized to their methyl esters according to Funada et al. with modifications (*Funada and Hirata, 1999*). Briefly the residues were dissolved in 1 ml 1 M sulfuric acid in methanol. For esterification the Falcon tubes were placed in an ultrasonic bath for 30 min. The fatty acid methyl esters (FAMEs) were extracted after addition of 3.3 ml water and 1.7 ml hexane by vigorous shaking on a Vortex for 1 min. The upper organic phase was withdrawn and dried over sodium carbonate. An aliquote was directy used for GC-MS analysis. A 1 mM fatty acid mixture in methanol (C$_{10:0}$, C$_{12:0}$, C$_{14:0}$, C$_{16:0}$, C$_{18:0}$, C$_{17:0}$ cyc (9,10), C$_{18:1}$ cis-Δ$^9$, C$_{18:1}$ trans-Δ$^9$, C$_{18:1}$ trans-Δ$^{11}$, C$_{18:2}$ cis,cis-Δ$^{9,12}$, C$_{18:2}$ trans,trans-Δ9,12 and C$_{18:3}$ cis,cis,cis-Δ$^{9,12,15}$) was diluted to 50, 100, 200 and 400 µM and derivatized in the same manner as above. The Agilent GC-MS system consisted of a gas chromatograph 7890A and an autosampler G4513A coupled to a quadrupole mass spectrometer MS G3172A (Agilent, CA, USA). Analytes were separated on a SGE BPX70 column (30 m x 0.32 mm i.d., 0.25 µm film thickness, Thermo Fisher Scientific, USA). Helium was used as carrier gas at a constant gas flow of 1.5 ml/min. The oven temperature program employed for analysis of FAMEs was as follows: 120°C; 20°C/min to 160°C; 3°C/min to 200°C; 20°C to 220°C, held for 8.7 min. The injector temperature was held at 250°C, and all injections (1 µl) were made in the split mode (1:10). The mass spectrometer was used in the electron impact (EI) mode at an ionizing voltage of 70 eV. Analytes were scanned over the range m/z 50 - 400 with a spectrum recording interval of 4 scans/sec. The GC interface temperature was held at 250°C. The MS source and quadrupole temperatures were kept at 280°C and 150°C, respectively. Data processing was performed by use of the software ChemStation E.02.02.1431 (Agilent, CA, USA). Fatty acids from PlaF samples were identified by comparison of their retention times and mass spectra with those of fatty acid standards and published data (*Yang et al., 2013*; *Benamara et al., 2014*; *Chao et al., 2010*). Quantification of FAMEs C$_{16:0}$ (1), C$_{17:0}$ cyc(9,10) (4), C$_{18:0}$ (5) and C$_{18:1}$ trans-Δ$^{11}$ (6) (*Figure 1*) were performed by external calibration with the corresponding reference compounds. C18:1 cis-Δ$^{11}$ (7) was quantified by use of the calibration curve of oleic acid (C$_{18:1}$ cis-Δ$^9$) justified by the almost congruent calibration curves of elaidic acid (C$_{18:1}$ trans-Δ$^9$) and C$_{18:1}$ trans-Δ$^{11}$.

## Crystallization, data collection, structure determination, and analysis

PlaF purified with OG was crystallized as described previously (*Bleffert et al., 2019*). The X-ray diffraction data were recorded at beamline ID29 of the European Synchrotron Radiation Facility (ESRF, Grenoble, France) and processed as described (*Bleffert et al., 2019*). The structure was determined by molecular replacement using the automated pipeline 'MrBUMP' from the CCP4 package (*Keegan et al., 2011*). In detail, a combination of PHASER (*McCoy et al., 2007*), REFMAC (*Murshudov et al., 1997*), BUCCANEER (*Cowtan, 2006*), and SHELXE (*Hübschle et al., 2011*) resulted in an interpretable electron density map to expand the placed model by molecular replacement using the model built with HsaD from *Mycobacterium tuberculosis* (PDB code: 2VF2) (*Lack et al., 2010*). Phase

improvement was achieved by running several cycles of automated model building (ARP/wARP, CCP4) and refinement using the PHENIX (*Adams et al., 2011*) package. The model was further corrected by manual rebuilding using the program COOT (*Emsley and Cowtan, 2004*). Detailed statistics on data collection and refinement are provided in *Table 1*. None of the residues is in disallowed regions according to Ramachandran plots generated with MolProbity (PHENIX) (*Adams et al., 2002*). The secondary structure was defined according to Kabsch and Sander (*Kabsch and Sander, 1983*). Interaction surface area was determined by PISA server (*Krissinel and Henrick, 2007*). Coordinates and structure factors for PlaF have been deposited in the Protein Data Bank under accession code 6I8W.

## Identification of structural homologs of PlaF

PlaF structural homologs were defined as protein structures from a non-redundant subset of PDB structures with less than 90% sequence identity to each other (PDB90 database, 12.10.2020) with a Z-score higher than 2 according to the DALI server (*Holm and Rosenström, 2010*). Sequence alignment based on structural superimposition of all 357 homologs of PlaF$_B$ (all 340 homologs of PlaF$_A$ were among PlaF$_B$ homologs) was used to identify proteins with homology in TM-JM helix of PlaF (residues 1–38). To evaluate homology, 39 3D structures with partial conservation of TM-JM helix were superimposed with the PlaF structure using Pymol (http://www.pymol.org) (*Figure 4—figure supplement 3*).

## Sequence analysis

A protein sequence of PlaF was used for a BLAST search of Pseudomonas Genome Databank (https://www.pseudomonas.com/) to identify PlaF orthologs in 4660 sequenced *P. aeruginosa* genomes. Pseudomonas Genome Databank BLAST search was extended to all pathogenic *Pseudomonas* species designated as those with assigned risk group 2 according to the German classification of prokaryotes into risk groups. NCBI BLAST (https://blast.ncbi.nlm.nih.gov/Blast.cgi) was used to identify PlaF homologs in other pathogenic bacteria.

## Molecular dynamics simulations of dimer and monomers

The crystal structure of the PlaF dimer was used as the starting point for building the systems for molecular dynamics (MD) simulations. Five missing C-terminal residues on both chains were added by using MODELLER (*Sali and Blundell, 1993*), and all small-molecule ligands were removed. The dimer was oriented into the membrane using the PPM server (*Lomize et al., 2012*). From the so-oriented dimer structure, chain B was deleted to obtain a PlaF$_A$ monomer in a dimer-oriented configuration; in the same way, chain A was deleted to keep PlaF$_B$. Additionally, the PlaF$_A$ and PlaF$_B$ monomers were oriented by themselves using the PPM server, yielding tilted configurations (t-PlaF$_A$ and t-PlaF$_B$). These five starting configurations, di-PlaF, PlaF$_A$, PlaF$_B$, t-PlaF$_A$, and t-PlaF$_B$, were embedded into a DOPE:DOPG=3:1 membrane with CHARMM-GUI v1.9 (*Jo et al., 2009*) resembling the native inner membrane of Gram-negative bacteria (*Benamara et al., 2014*; *Murzyn et al., 2005*). A distance of at least 15 Å between the protein or membrane and the solvation box boundaries was considered. KCl at a concentration of 0.15 M was included in the solvation box to obtain a neutral system. The GPU particle mesh Ewald implementation from the AMBER16 molecular simulation suite (*Le Grand et al., 2013*; *Darden et al., 1993*) with the ff14SB (*Maier et al., 2015*) and Lipid17 (*Dickson et al., 2014*; *Skjevik et al., 2016*; *Case, 2017*) parameters for the protein and the membrane lipids, respectively, were used; water molecules were added using the TIP3P model (*Jorgensen et al., 1983*). For each protein configuration, 10 independent MD simulations of 2 μs length were performed. Covalent bonds to hydrogens were constrained with the SHAKE algorithm (*Ryckaert et al., 1977*) in all simulations, allowing the use of a time step of 2 fs. Details of the thermalization of the simulation systems are given below. All unbiased simulations showed stable protein structures (*Figure 7—figure supplement 1*) and membrane phases, evidenced by electron density and order parameter calculations (*Figure 7—figure supplement 1*). The area per lipid through all simulations calculated for the leaflet opposite to the one where PlaF was embedded was 61.3±0.13 Å$^2$ (mean ± SEM), similar to values reported previously (*Murzyn et al., 2005*).

### Thermalization and relaxation of simulated systems

Initially, systems were energy-minimized by three mixed steepest descent/conjugate gradient calculations with a maximum of 20,000 steps each. First, the initial positions of the protein and membrane were restrained, followed by a calculation with restraints on the protein atoms only, and finally a minimization without restraints. The temperature was maintained by using a Langevin thermostat (*Quigley and Probert, 2004*), with a friction coefficient of 1 ps⁻¹. The pressure, when required, was maintained using a semi-isotropic Berendsen barostat (*Berendsen et al., 1984*), coupling the membrane (x-y) plane. The thermalization was started from the minimized structure, which was heated by gradually increasing the temperature from 10 to 100K for 5 ps under NVT conditions, and from 100 to 300K for 115 ps under NPT conditions at 1 bar. The equilibration process was continued for 5 ns under NPT conditions, after which production runs were started using the same conditions.

### Structural analysis of MD trajectories

All analyses were performed by using CPPTRAJ (*Roe and Cheatham, 2013*). The distance between the centers of mass (COM) of residues 25–38 $C_\alpha$ atoms of the chains in the dimer structure was evaluated (*Figure 7—figure supplement 1*); this residue range corresponds to the solvent-accessible half of helix TM-JM (*Figures 7a and 8*). For the monomer structures, the angle with respect to the membrane normal was assessed. For this, the angle between the membrane normal and the vector between the COM of residues 21–25 and residues 35–38 was calculated (*Figure 7b*).

### PMF and free energy calculations of dimer dissociation

For calculating a configurational free energy profile (PMF) of the process of dimer dissociation, 36 intermediate states were generated by separating one chain of the dimer along the membrane plane by 1 Å steps, resulting in a minimum and maximum distance between the chain COM of 40.8 and 68 Å, respectively. The generated structures represent the separation process of the PlaF dimer. To sample configurations along the chain separation in a membrane environment, each intermediate state was embedded into a membrane of approximately 157×157 Å² by using PACKMOL-Memgen (*Schott-Verdugo and Gohlke, 2019*), and independent MD simulations of 300 ns length each, with a total simulation time of 10.8 µs. Umbrella sampling simulations were performed by restraining the initial distance between chains in every window with a harmonic potential, using a force constant of 4 kcal mol⁻¹ Å⁻² (*Torrie and Valleau, 1977*); the distance between the COM of $C_\alpha$ atoms of residues 25–38 of each monomer was used as a reaction coordinate, being restrained in every simulation. Values for the reaction coordinate, representing the intermonomer distance *r*, were recorded every 2 ps and post-processed with the Weighted Histogram Analysis Method implementation of A. Grossfield (WHAM 2.0.9) (*Suzuki, 1975*; *Grossfield, 2016*), removing the first 100 ns as an equilibration of the system. The kernel densities showed a median overlap of 8.2% between contiguous windows (*Figure 7—figure supplement 1*), well suited for PMF calculations (*Chen and Kuyucak, 2011*). The error was estimated by separating the last 200 ns of data in four independent parts of 50 ns each and then calculating the standard error of the mean of the independently determined energy profiles.

The association free energy was estimated from the obtained PMF following the membrane two-body derivation from *Johnston et al., 2012* and our previous work (*Pagani and Gohlke, 2018*). The PMF of dimer association is integrated along the reaction coordinate to calculate an association constant ($K_a$), which is transformed to the mole fraction scale ($K_x$) taking into account the number of lipids $N_L$ per surface area $A$, and this value is used to calculate the difference in free energy between dimer and monomers ($\Delta G$), according to *Equations 1–3*:

$$K_a = \frac{||\Omega||}{(2\pi)^2} \int_0^D r e^{\frac{-w(r)}{k_B T}} dr \tag{1}$$

$$K_x = K_a \frac{N_L}{A} \tag{2}$$

$$\Delta G = -RT \ln\left(K_x\right) \tag{3}$$

where *r* is the value of the reaction coordinate, *w(r)* is the PMF at value *r*, *D* is the maximum distance at which the protein is still considered a dimer, $k_B$ is the Boltzmann constant, and *T* is the temperature at which the simulations were performed. The factor $\frac{||\Omega||}{(2\pi)^2}$ considers the restriction of the configurational

space of the monomers upon dimer formation in terms of the sampled angle between the two chains in the dimeric state (*Equation 4*) and the accessible space for the monomers, $(2\pi)^2$.

$$||\Omega|| = \left[\max(\theta_a) - \min(\theta_a)\right] * \left[\max(\theta_b) - \min(\theta_b)\right] \tag{4}$$

In *Equation 4*, the angle $\theta_a$ is defined as the angle formed between the vectors connecting the COM of chain $b$ with the COM of the chain $a$ and with the COM of residues 25–38 of the latter chain; $\theta_b$ is defined analogously starting from the COM of chain $a$. A value for $||\Omega||$ of 0.55 computed from *Equation 4* indicates the fraction of the accessible space that the PlaF monomers have in the dimeric state compared to when both chains rotate independently $[(2\pi)^2]$.

## PMF and free energy calculations of monomer tilting

The initial conformations used in every window for calculating the PMF of the monomer tilting were obtained from the first microsecond of MD simulations of replica 10 of $PlaF_A$ (oriented as in the di-PlaF crystal structure) where spontaneous tilting occurred. The distance $d$ along the z-axis between the COM of $C_\alpha$ atoms of residues 33–37 of the monomer with the membrane center was used to select 22 intermediate tilting configurations. $d$ significantly correlates ($R^2$=0.997, p<0.001) with the angle formed by the second half of helix αJM1 of the monomer (residues 25–38) and the normal vector of the membrane (*Figure 7—figure supplement 1*). The starting conformations were extracted from the representative trajectory, taking the respective snapshots where $d$ and the angle showed the least absolute deviation to the average value obtained by binning $d$ in windows of 2 Å width and with an evenly distributed separation of 1 Å. The distance $d$ was restrained for every configuration by a harmonic potential with a force constant of 4 kcal mol$^{-1}$ Å$^{-2}$, and sampling was performed for 300 ns per window. The data were obtained every 2 ps and analyzed as described above, resulting in 8.6% of median overlap between kernel densities of contiguous windows (*Figure 7—figure supplement 1*). The error was estimated in the same way as for the dimerization (see above).

For calculating the free energy difference between the obtained basins, the PMF of monomer tilting was integrated using *Equations 5 and 6* (*Doudou et al., 2009*):

$$K_{tilting} = \frac{\int_{B_1} e^{-\frac{w(d)}{k_B T}} dr}{\int_{B_2} e^{-\frac{w(d)}{k_B T}} dr} \tag{5}$$

$$\Delta G_{tilting} = -RT \ln K_{tilting} \tag{6}$$

where $d$ is defined as above, $w(d)$ is the value of the PMF at that distance, and $B_1$ and $B_2$ represent the basins for the tilted and split configurations, respectively. The integration limits $B_1$ and $B_2$ included each basin portion below half of the value between the basin minimum and the energy barrier separating the basins, respectively (*Figure 7c*, yellow shaded regions).

## PlaF dimer versus monomer proportion under in vivo conditions

The dimer to monomer equilibrium of PlaF in the membrane results from the coupling of the following equilibria:

$$2M \overset{K_a}{\rightleftharpoons} D \qquad K_a = \frac{[D]}{[M]^2} \tag{7}$$

$$M \overset{K_{tilting}}{\rightleftharpoons} M_{tilted} \qquad K_{tilting} = \frac{[M_{tilted}]}{[M]} \tag{8}$$

yielding,

$$D \overset{K_a K_{tilting}^{-2}}{\rightleftharpoons} 2M_{tilted} \tag{9}$$

where D, M, and $M_{tilted}$ represent the PlaF dimer, 'split' monomer, and tilted monomer, respectively, with $K_a$ and $K_{tilting}$ being the dimer association and monomer tilting equilibrium constants, obtained from the PMF calculations. Based on the association constant computed according to *Equation 7*, $K_a$=[D]/[M]$^2$=1.57×10$^7$ Å$^2$, with [D] and [M] as area concentrations of dimer and monomer, respectively, the proportion of PlaF dimer versus monomer in a live cell *of P. aeruginosa* can be computed.

Experimentally, 40 µg GPLs per 1 ml *P. aeruginosa* p-*plaF* (OD$_{580nm}$ 1) were extracted, and a PlaF purification yield of ca. 1 µg from 1 ml *P. aeruginosa* p-*plaF* culture with OD$_{580nm}$ was obtained (*Bleffert et al., 2019*; *Supplementary file 3*). Considering the molecular weight of PlaF of 35.5 kDa and assuming 750 Da as the average molecular weight of membrane GPL, this relates to a concentration under overexpressing conditions of ~$5.28 \times 10^{-4}$ PlaF monomers per lipid. Under non-overexpressing conditions, the concentration of PlaF monomers is estimated to be at least 100- to 1000-fold lower, that is, $5.28 \times 10^{-6}$ to $5.28 \times 10^{-7}$ PlaF monomers per lipid. Considering that the area per lipid in a PE:PG=3:1 membrane at 300K is approximately 61 Å$^2$ per leaflet (or 30.5 Å$^2$ in a bilayer, computed in this work and *Murzyn et al., 2005*), the total area concentration of PlaF molecules then is

$$T \;=\; 2\left[D\right] + \left[M\right] = \left[1.73 \text{ x } 10^{-8}, \; 1.73 \text{ x } 10^{-7}\right] \frac{\text{PlaF}}{\text{Å}^2}. \tag{10}$$

Expressing the association constant in terms of the monomer concentration using *Equation 7* yields

$$K_a = \frac{\frac{T-[M]}{2}}{[M]^2} \qquad \Leftrightarrow \qquad 2 K_a \left[M\right]^2 + \left[M\right] - T = 0, \tag{11}$$

and solving the quadratic equation then results in

$$\left[M\right] = \frac{-1+\sqrt{1+8K_a T}}{4 K_a} = \left[1.25 \times 10^{-8}, 6.00 \times 10^{-8}\right] \frac{\text{PlaF}}{2} \tag{12}$$

and

$$\left[D\right] = \frac{T-[M]}{2} = \left[2.43 \times 10^{-9}, 5.66 \times 10^{-8}\right] \frac{\text{PlaF dimer}}{2}, \tag{13}$$

These results show that in live cells, the fraction of PlaF in the monomeric (dimeric) state is between 35% and 72% (65% and 28%), where the PlaF monomer is considered to be in the 'split' configuration with respect to the membrane normal.

As the tilting of the PlaF monomer is energetically favorable compared to the 'split' configuration and, hence, depletes the concentration of 'split' PlaF monomers, the dimeric PlaF concentration will decrease (*Figure 7a*). To quantitatively consider the effect of the tilting, we express the overall equilibrium constant for the processes shown in *Figures 7a and 8*, and described in *Equations 7–9* as

$$K = K_a \, K_{tilting}^{-2} = \frac{[D]}{\left[M_{tilted}\right]^2}, \tag{14}$$

where

$K_{tilting} = \frac{[M_{tilted}]}{[M]} = 3.35$, equivalent to $G_{\text{tilting}} = -0.72 \frac{\text{kcal}}{\text{mol}}$, computed according to *Equation 5*.

Following the same procedure as before then yields

$$\left[M_{tilted}\right] = \left[1.66 \times 10^{-8}, 1.28 \times 10^{-7}\right] \frac{\text{PlaF}}{2}$$

$$\left[D\right] = \left[3.83 \times 10^{-10}, 2.28 \times 10^{-8}\right] \frac{\text{PlaF dimer}}{2},$$

showing that in live cells, the fraction of PlaF in the tilted monomeric (dimeric) state is between 74% and 96% (26 and 4%). A graphical representation of the percentage of protein as a tilted monomer with respect to the protein concentration in the membrane is shown in *Figure 7e*.

## Acknowledgements

This study was funded by the Deutsche Forschungsgemeinschaft (DFG, German Research Foundation), Project CRC 1208 (number 267205415) to FK and KEJ (subproject A02), and HG (subproject A03). The authors are grateful to the beamline scientists at the European Synchrotron Radiation Facility (Grenoble, France) for assisting with the use of beamline ID29. The authors thank R Voulhoux (CNRS AMU LCB, Marseille) for providing anti-SecG antiserum, P Dollinger (HHU Düsseldorf) for help with MST measurements, M Modri (HHU Düsseldorf) for help with PG_15:0_18:1 assay, and M Dick (HHU Düsseldorf) for assistance in setting up biased MD simulations. The authors are grateful for computational support by the 'Zentrum für Informations und Medientechnologie' at the

Heinrich-Heine-Universität Düsseldorf and the computing time provided by the John von Neumann Institute for Computing (NIC) to HG on the supercomputer JUWELS at Jülich Supercomputing Centre (JSC) (user IDs: HKF7; VSK33; HDD18; plaf).

## Additional information

### Funding

| Funder | Grant reference number | Author |
|---|---|---|
| Deutsche Forschungsgemeinschaft | 267205415 | Holger Gohlke<br>Karl-Erich Jaeger<br>Filip Kovacic |
| European Synchrotron Radiation Facility | | Renu Batra-Safferling |

The funders had no role in study design, data collection and interpretation, or the decision to submit the work for publication.

### Author contributions

Florian Bleffert, Stephan N Schott-Verdugo, Meike Siebers, Björn Thiele, Formal analysis, Investigation, Validation, Writing – review and editing; Joachim Granzin, Data curation, Formal analysis, Investigation, Validation, Writing – review and editing; Muttalip Caliskan, Formal analysis, Investigation, Writing – review and editing; Laurence Rahme, Peter Dörmann, Supervision, Validation, Writing – review and editing; Sebastian Felgner, Formal analysis, Investigation, Validation, Visualization, Writing – review and editing; Holger Gohlke, Conceptualization, Funding acquisition, Supervision, Validation, Writing - original draft, Writing – review and editing; Renu Batra-Safferling, Conceptualization, Formal analysis, Investigation, Validation, Visualization, Writing - original draft, Writing – review and editing; Karl-Erich Jaeger, Funding acquisition, Project administration, Supervision, Writing – review and editing; Filip Kovacic, Conceptualization, Data curation, Formal analysis, Funding acquisition, Investigation, Methodology, Project administration, Resources, Supervision, Validation, Visualization, Writing - original draft, Writing – review and editing

### Author ORCIDs

Laurence Rahme http://orcid.org/0000-0002-5374-4332
Sebastian Felgner http://orcid.org/0000-0003-0030-2490
Peter Dörmann http://orcid.org/0000-0002-5845-9370
Holger Gohlke http://orcid.org/0000-0001-8613-1447
Renu Batra-Safferling http://orcid.org/0000-0002-8597-4335
Filip Kovacic http://orcid.org/0000-0002-0313-427X

### Decision letter and Author response

Decision letter https://doi.org/10.7554/eLife.72824.sa1
Author response https://doi.org/10.7554/eLife.72824.sa2

## Additional files

### Supplementary files

• Supplementary file 1. All phospholipid species identified in P. aeruginosa PA01 and ΔplaF by Q-TOF MS/MS.

• Supplementary file 2. Phospholipid species significantly differentially abundant in P. aeruginosa wild-type, ΔplaF, and ΔplaF::plaF.

• Supplementary file 3. Properties of the cultures used for lipid extraction.

• Supplementary file 4. PlaF homologs in P. aeruginosa species with sequenced genomes.

• Supplementary file 5. List of interactions involving the ligand molecules.

• Supplementary file 6. List of interactions involving the dimer interface.

- Supplementary file 7. List of interactions# involving the catalytic triad residues S137, D258 and H286.
- Supplementary file 8. Residues lining the active site cavity and their interactions with ligands.
- Supplementary file 9. Michaelis-Menten constants for inhibition of PlaF with decanoic acid (FA C10).
- Supplementary file 10. Average 2D-RMSDall atom of residues 25 to 315 of the structures sampled along MD trajectories.[a].
- Transparent reporting form
- Source data 1. All supplementary data in editable format.

### Data availability

Diffraction data have been deposited in PDB under the accession code 6I8W. All data generated or analysed during this study are included in the manuscript and supporting file. Sequencing data are embedded in Fig. S1b. Source Data file "Table S1 - lipidome" has been provided for Figure 2. It contains the numerical data used to generate the figure 2c. Source data used to calculate the potentials of mean force and their corresponding simulation trajectory files shown in Figure 7 and Figure 7—figure supplement 1 are accessible at the DSpace instance researchdata.hhu.de under DOI:http://doi.org/10.25838/d5p-31.

The following datasets were generated:

| Author(s) | Year | Dataset title | Dataset URL | Database and Identifier |
|---|---|---|---|---|
| Granzin J, Batra-Safferling R | 2019 | Crystal structure of a membrane phospholipase A, a novel bacterial virulence factor | https://www.rcsb.org/structure/6I8W | RCSB Protein Data Bank, 6I8W |
| Schott-Verdugo S, Gohlke H, Batra-Safferling R, Jaeger KE, Kovacic F | 2022 | Structural and mechanistic insights into bacterial phospholipase A involved in membrane phospholipid degradation and virulence | http://doi.org/10.25838/d5p-31 | HHU ResearchData, 10.25838/d5p-31 |

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
