## [Editor Report]

This study provides new insights into how a bacterial phospholipase called PlaF degrades membrane phospholipids in a controlled fashion to allow bacteria to alter their membrane composition to adapt to changing conditions. Inas much as PlaF is important for virulence, it will be interesting to see if the comprehensive biochemical and structural analysis in the current paper will aid in the development of a class of antibiotics targeting PlaF.

---

## [Decision Letter]

**Decision letter after peer review:**

Thank you for submitting your article "Evidence for a bacterial Lands cycle phospholipase A involved in membrane phospholipid remodeling" for consideration by *eLife*. Your article has been reviewed by 2 peer reviewers, and the evaluation has been overseen by a Reviewing Editor and Olga Boudker as the Senior Editor. The following individual involved in review of your submission has agreed to reveal their identity: Lei Zheng (Reviewer #1).

This study provides new insights into the mechanisms and regulation of phospholipid remodeling in bacteria by a phospholipase PlaF. The reviewers were enthusiastic about the several new findings in this study and have provided several suggestions to address in a revised submission. The major concern, which is not necessarily a negative, is that the manuscript includes a massive amount of data and the connection is at times hard to follow. The strengths of the manuscript are the comprehensive biochemical and structural analysis of how PlaF functions, whereas some of the other data are in comparison, more preliminary in nature. The authors should consider focusing the manuscript (including the title) on these strengths.

Please see specific points to address in the reviews from both reviewers below.

Other Specific Comments from Reviewer #1:

You have included a massive amount of data in the manuscript. The biochemical and structural data are really nice! Other data are interesting, but preliminary, which may dilute the novelty of your work. I would suggest focusing on the molecular mechanism of PlaF, which may help sharpen the paper.

Other Specific Comments from Reviewer #3:

In addition to add experiments as above, it is important to rearrange the manuscript for more clear and smooth flow. It is also advisable to describe more accurately what is known on Lands cycle in mammals.

*Reviewer #1 (Recommendations for the authors):*

In this manuscript, Kovacic et al., report structure and functional studies of PlaF from *Pseudomonas aeruginosa*. They found that PlaF functions as a novel phospholipase A in the bacterial inner membrane with its catalytic domain facing towards the periplasm. They determined the X-ray structure of PlaF to 2A resolution. The structure shows a dimeric conformation with free fatty acid (FFA) molecules bound in the substrate-binding pocket. Based on the structure, they performed very nice studies using biochemical assays and MD simulation to propose a regulatory mechanism of PlaF via transmembrane helix-mediated dimerization. In general, this is a very interesting and comprehensive study. The works provide new insights into the GPL homeostasis on the bacterial cell membrane. The conclusion is supported by most of the data, except some issues as follow:

1. The structure looks well-resolved based on the statistics. However, the electron density of FFA in Chain B is very weak, barely seen at 1 σ. Does FFA have different occupancy between Chain A and Chain B? Does this variation contribute to the conformational changes between the two monomers?

2. The hypothesis of GPL remodeling in Figure 2d is too speculative and is not supported by their data in Figure 2c. Is it possible the changes of GPLs are caused by the depletion of PlaF on the membrane?

3. A similar concern regarding Figure 3. The phenotype of the plaF KO strain may not establish the PlaF protein as an antibacterial target. Is the susceptibility caused by the loss of PlaF catalytic activity or simply due to the depletion of plaF? Authors need to be cautious when discussing potential drug binding sites based on the structure in the discussion.

4. It is unclear how FFA binding facilitates PlaF dimerization. It would be important to discuss it based on the structure and biochemical data.

5. It is unclear any ligands have been included in the MD simulation. FFA or OG?

6. The title is vague and a better one may help highlight the major findings of this work.*Reviewer #3 (Recommendations for the authors):*

The work consists of three parts, 1)phospholipase activity, possibly involved in remodeling pathway of bacterial GPL, 2) structural insight of fatty acid-induced dimeric form with weak catalytic activity, and 3) virulent nature of *Pseudomonas* against insect and bone marrow-derived macrophages. All are new findings, but the relation of three topics is not wrriten clear enough, and each point needs further detailed analyses as follows.

1. For the phospholipase activity, authors have better determine enzyme activities for individual GPL with different fatty acid species using LC-MS with appropriate internal standards. It has to be shown more clearly whether only sn-1 is cleaved, but not sn-2, and whether PLaF does not have transacylase activity. It is also critical to make several mutants of active site, and measure both carboxyesterase and PLA activities in parallel.

2. It is not clear whether dimer formation is just an artifact by over-expression. How PLaFa and PlaFb are formed? What is biological significance of fatty acid effect at these high concentrations? Is it not detergent effects with certain HLB?

3. The mechanism of virulent effect has to be more clearly determined. Does it relate to phospholipase, or caroboxyesterase, or both? Does PlaF induce membrane lysis of macrophages or insect cells by producting lysoGPL?. More mechanical insights are necessary.

4. Discussion is necessary how the remodeling enzyme plays also virulent roles.

---

## [Author Response]

This study provides new insights into the mechanisms and regulation of phospholipid remodeling in bacteria by a phospholipase PlaF. The reviewers were enthusiastic about the several new findings in this study and have provided several suggestions to address in a revised submission. The major concern, which is not necessarily a negative, is that the manuscript includes a massive amount of data and the connection is at times hard to follow. The strengths of the manuscript are the comprehensive biochemical and structural analysis of how PlaF functions, whereas some of the other data are in comparison, more preliminary in nature. The authors should consider focusing the manuscript (including the title) on these strengths.Please see specific points to address in the reviews from both reviewers below.Other Specific Comments from Reviewer #1:You have included a massive amount of data in the manuscript. The biochemical and structural data are really nice! Other data are interesting, but preliminary, which may dilute the novelty of your work. I would suggest focusing on the molecular mechanism of PlaF, which may help sharpen the paper.Other Specific Comments from Reviewer #3:In addition to add experiments as above, it is important to rearrange the manuscript for more clear and smooth flow. It is also advisable to describe more accurately what is known on Lands cycle in mammals.

We have thoroughly revised the manuscript as suggested by the reviewers. In particular, the title was modified, the introduction extended, the discussion shortened and focused on the main results. Additional information on the Lands cycle in eukaryotes is now provided in the introduction. Please note that the first structure of a lysophospholipid acyltransferase (LPLAT, Zhang, Q., *et al.*, The structural basis for the phospholipid remodeling by lysophosphatidylcholine acyltransferase 3. Nat Commun, 2021. 12(1): p. 6869.) and classification of LPLATs (Valentine, W.J., *et al.*, Update and nomenclature proposal for mammalian lysophospholipid acyltransferases, which create membrane phospholipid diversity. Journal of Biological Chemistry, 2022. 298(1): p. 101470.) were published during the revision of this manuscript. Furthermore, despite many known mammalian PLAs, the relationship of these enzymes and Lands cycle is still not well understood.

Reviewer #1 (Recommendations for the authors):In this manuscript, Kovacic et al., report structure and functional studies of PlaF from *Pseudomonas aeruginosa*. They found that PlaF functions as a novel phospholipase A in the bacterial inner membrane with its catalytic domain facing towards the periplasm. They determined the X-ray structure of PlaF to 2A resolution. The structure shows a dimeric conformation with free fatty acid (FFA) molecules bound in the substrate-binding pocket. Based on the structure, they performed very nice studies using biochemical assays and MD simulation to propose a regulatory mechanism of PlaF via transmembrane helix-mediated dimerization. In general, this is a very interesting and comprehensive study. The works provide new insights into the GPL homeostasis on the bacterial cell membrane. The conclusion is supported by most of the data, except some issues as follow:1. The structure looks well-resolved based on the statistics. However, the electron density of FFA in Chain B is very weak, barely seen at 1 σ. Does FFA have different occupancy between Chain A and Chain B? Does this variation contribute to the conformational changes between the two monomers?

Thank you for pointing this out. The electron density of FFA in chain B is indeed less well-defined than that of FFA in chain A. However, we believe that the confusion here is due to the calculated electron density maps provided by the PDB, which look very weak around the region of the FFA in molecule B compared to the electron density maps we obtained directly from the Phenix refinement program, which we have used for the interpretation. The main reason for the discrepancies between our maps and those from the PDB is that different programs are used for bulk-solvent correction and overall scaling during refinement. We will gladly provide the MTZ file with the original phases at the request of the reviewer.

No occupancy was refined for the FFA molecules, but the overall B-factors show significant differences in comparison. For the UNDECANOIC ACID of chain B, the average B factor is 89.03 Å2 (average B factor of the surrounding protein chain is 49.29 Å2) in contrast to the MYRISTIC ACID of chain A which has an average B factor of 66.63 Å2 (average B factor of the surrounding protein chain: 47.61 Å2). The average B factor thus shows that the electron density map of FFA(B) is considerably poorly defined, i.e., the FFA has a high flexibility or lower occupancy, as pointed out by the reviewer. We cannot say with certainty or rule out the possibility that these variations are responsible for the differences between the two protein molecules since the overall structure is also influenced in part by crystal packing effects.

We have added this information in the results on page 7 (lanes 10-12).

2. The hypothesis of GPL remodeling in Figure 2d is too speculative and is not supported by their data in Figure 2c. Is it possible the changes of GPLs are caused by the depletion of PlaF on the membrane?

We have now modified figure 2 by removing the speculative part about the putative Lands cycle.

If we keep in mind that PlaF is present in *P. aeruginosa* cells at very low amounts (below the detection limit of a Western blot, see Figure 7-supplementary figure 2), it is unlikely that such considerably large changes of GPL content are due to the absence of PlaF. However, it cannot be excluded that the absence of PlaF in the cytoplasmic membrane of *P. aeruginosa* Δ*plaF* is responsible for the observed GPL changes. We have now commented on that in the discussion (page 12, lanes 28-30).

Our model predicts that PlaF which is anchored to the cytoplasmic membrane modifies this membrane by hydrolysing GPLs. This appears plausible and in agreement with a) in vitro catalytic activity of PlaF against a range of cytoplasmic GPLs (Figure 2), and b) decreased PLA activity of the Δ*plaF* strain compared to the wild type (Figurre 2-supplementary figure 2).

3. A similar concern regarding Figure 3. The phenotype of the plaF KO strain may not establish the PlaF protein as an antibacterial target. Is the susceptibility caused by the loss of PlaF catalytic activity or simply due to the depletion of plaF? Authors need to be cautious when discussing potential drug binding sites based on the structure in the discussion.

We have now removed a large part of the discussion about ligand binding sites of PlaF as potential drug binding sites.

We are aware that further experimental evidence is needed to establish PlaF as an antibacterial target and these experiments are ongoing. Our preliminary results point to the importance of PlaF as a virulence regulator in *P. aeruginosa*. We observed that PlaF strongly affects virulence-related phenotypes, namely biofilm production and architecture (Author response image 1) and swimming motility (Author response image 2). Additionally, we observed an extended lag-phase during the growth of *P. aeruginosa* Δ*plaF* compared with the wild-type, which may be explained by impaired iron transport in Δ*plaF* (Author response image 3).

**Author response image 1. sa2fig1:** PlaF is a novel *P. aeruginosa* biofilm effector. (**A**) Biofilm amount of *P. aeruginosa* Δ*plaF* and the wild-type strain cultivated in 96-well MTP (LB medium, 37°C, without aeration) was quantified by staining the cells attached to MTP with crystal violet. The results are mean ± standard deviation of five biological replicates, each measured eight times. Statistical analysis was performed using the *t*-test, * p < 0.05. (**B**) Biofilm architecture was analysed by CLSM after 24, 72, and 144 h growth at 37ºC in a flow cell with a continuous supply of LB medium. Shown are representative figures of two biological replicates analysed by imaging three different sections (100 x 100 µm) of the cover glass.

**Author response image 2. sa2fig2:** *P. aeruginosa* Δ*plaF* mutant strain is impaired in swimming motility. The growth of the strains on LB agar swimming plates was quantified using ImageJ and expressed as swimming area, *** *p* < 0.00005, *n* = 10.

**Author response image 3. sa2fig3:** Iron-acquisition is disbalanced in *P. aeruginosa* ∆*plaF*. A) Strains were grown statically at 37°C in 96-well microtiter plates for biofilm formation or in Erlenmeyer flasks under planktonic conditions for 24 h. Pyoverdine concentration was quantified in cell-free supernatant of *P. aeruginosa* PA01 and ∆*plaF* grown in plastic 96-well microtiter plates (biofilm) or in Erlenmeyer flasks (planktonic) at 37°C for 24 h. Results are mean ± S.D. B) *P. aeruginosa* ΔplaF shows a lower intrecellular iron concentration than the wild type. Iron was quantified [Tamarit, J., *et al.*, Anal Biochem, 2006] in cells isolated from the cultures grown 18 h in LB medium at 37 C. C) The growth curves of *P. aeruginosa* PA01 and ∆*plaF* (n = 3) in M9 minimal medium under iron-limiting conditions and in the presence of 100 µM FeSO_4_ were determined by measuring the optical density at 580 nm. Results are mean ± S.D. The growth of *P. aeruginosa* ∆*plaF* and PA01, as well as *P. aeruginosa* ∆*plaF* and ∆*plaF* + FeSO_4_, differ significantly (p < 0.05).

To understand how PlaF modulates the virulence of *P. aeruginosa* on a molecular level, we have performed a global proteomic analysis of wild-type and Δ*plaF* strains. The results revealed a pleiotropic effect of PlaF deletion on many virulence-related pathways, among them attachment, signalling, iron homeostasis, chemotaxis (Figures 4 – 6). We suggest that changes in GPL composition modulate the function of cytoplasmic membrane proteins, which in turn triggers a cellular response. These still unpublished results are part of a manuscript in preparation for submission and can be provided to the reviewers upon request.

We decided not to include proteomics results in the *eLife* manuscript as the manuscript already contains many data as outlined by the reviewers.

4. It is unclear how FFA binding facilitates PlaF dimerization. It would be important to discuss it based on the structure and biochemical data.

We agree that the here presented dimeric structure of PlaF cannot provide a clear answer on how FFA binding facilitates PlaF dimerisation. FFA inhibition kinetics indicate that additionally to competitive inhibition (blocking of the active site by bound fatty acid), there is a non-competitive inhibition component which might implicate that FFAs trigger the formation of the inactive dimer.

Our observation that the dimer interface and the catalytic site are connected through bound FFA and a pseudo-ligand (octyl glucoside) provides a structural hint about the link of dimerisation, activity and FFA binding.

This was now briefly discussed on page 14, lanes 30-32.

5. It is unclear any ligands have been included in the MD simulation. FFA or OG?

We did not include any fatty acid or detergent in our simulations. They are shown in the figures to visualise the active site of PlaF and a putative path for substrate access. This information was now added in the caption of figure 7 and the M and M part.

6. The title is vague and a better one may help highlight the major findings of this work.

We now changed the title to highlight the structural and mechanistic results.

Reviewer #3 (Recommendations for the authors):The work consists of three parts, 1)phospholipase activity, possibly involved in remodeling pathway of bacterial GPL, 2) structural insight of fatty acid-induced dimeric form with weak catalytic activity, and 3) virulent nature of Pseudomonas against insect and bone marrow-derived macrophages. All are new findings, but the relation of three topics is not wrriten clear enough, and each point needs further detailed analyses as follows.1. For the phospholipase activity, authors have better determine enzyme activities for individual GPL with different fatty acid species using LC-MS with appropriate internal standards. It has to be shown more clearly whether only sn-1 is cleaved, but not sn-2, and whether PLaF does not have transacylase activity. It is also critical to make several mutants of active site, and measure both carboxyesterase and PLA activities in parallel.

Thank you for your suggestion. We have now performed an experiment in which 1-(9Z-octadecenoyl)-2-pentadecanoyl-glycero-3-phospho-(1'-rac-glycerol) (PG_15:0_18:1) was hydrolysed by PlaF following quantification of fatty acids by GC-MS. The experiment was challenging because of a large amount of purified PlaF needed.

In two independent experiments, 0.4 µM pentanoic acid was detected after hydrolysis by PlaF and FA extraction. The concentration of oleic acid (18:1) in PlaF-treated samples was not higher than in the blank samples. Oleic acid is bound at *sn*2 position in the lipid and pentanoic acid at *sn*1 position; thus, we can conclude that PlaF is a PLA1 as shown previously (see Figure 2a). We have now added these results as Figure 2- supplementary figure 1 and comment on them in the main text (page 5, lanes 7-9).

The colorimetric NEFA assay which we have used to measure PLA activity is a standard assay and was applied to characterize other PLAs [1, 2]. The advantage of the NEFA assay is its sensitivity and, in comparison to a GC-MS-based assay, the NEFA assay does not involve an error-prone fatty acid extraction step.

We have previously shown that the variant PlaF-S137A with the catalytic serine replaced by a non-functional alanine does not possess esterase activity [3]. Furthermore, a cell lysate from *P. aeruginosa* Δ*plaF* showed 40% lower esterase activity than that of the wild type strain (Author response image 4). As PlaF is a one-domain protein and has a catalytic triad conserved among many esterases and PLAs, we propose that identical residues are involved in PLA and esterase activities.

**Author response image 4. sa2fig4:** Esterase activity of *P. aeruginosa* Δ*plaF* strain. *P. aeruginosa* PAO1 and Δ*plaF* strains were grown overnight in LB-medium at 37°C. Cells were harvested by centrifugation, suspended in Tris‑HCl buffer (100 mM, pH 8) to equal cell count, and enzyme activities were determined with *p*-nitrophenyl butyrate as the substrate. The results are mean ± standard deviations of three biological replicates, each measured three times.

During the revision of this manuscript, we have published an article describing the putative mechanism of GPL hydrolysis by PlaF [4]. There, we report the results of a mutational analysis of the active site of PlaF, which lead us to suggest how GPL enters the active site pocket of PlaF and how the products are likely released to prepare PlaF for a new catalytic cycle.

At present, we do not have any indications as to an acyltransferase activity of PlaF. We regard the determination of this enzymatic activity and, if detected, of its physiological consequences as a subject of a follow-up study.

2. It is not clear whether dimer formation is just an artifact by over-expression. How PLaFa and PlaFb are formed? What is biological significance of fatty acid effect at these high concentrations? Is it not detergent effects with certain HLB?

We agree with the possibility that overexpression leads to dimer formation. Unfortunately, under laboratory conditions, PlaF is produced by *P. aeruginosa* wild type at low concentrations (Figure 7-supplementary figure 2); therefore, we have used mild overexpression (using the low copy number plasmid pBBR1mcs-3) to capture PlaF dimers in the cell. Our results suggest equilibrium between dimeric and monomericPlaF in the cell membrane. in vitro results indicate that this equilibrium is shifted towards dimer formation by raising the concentrations of either PlaF or fatty acids. Although we do not entirely understand the structural prerequisites for dimerisation, we know that PlaF forms dimers and dimerisation regulates its activity.

An in vitro effect of fatty acids on PlaF activity and dimerisation fits a product feedback mechanism which would make sense for regulation of PlaF, as it is potentially toxic to *P. aeruginosa*. Although precise data about the intracellular concentration of FFA in *P. aeruginosa* are not available, Fang L. and co-authors [5] recently published that the intracellular concentration of FFAs in *E. coli* is 0.5 – 1 mg/l (2 – 4 mM for palmitic acid). This is in the range of the concentrations used in our experiments.

The structure of di-PlaF in which a) hydrophobic residues of TM helices form intermolecular interactions, b) polar residues of juxtamembrane domains of two PlaF molecules interacts with each other, and c) charged residues are preceding the TM helix agrees with the topology suggested for several TM helix dimers (see figure 1 in reference [6]). Therefore, we rationalise that the PlaF dimer detected in the described crystal structure represents a physiological state.

The effects of detergents or membrane solubilising compounds on the structure of membrane proteins is the subject of a long-standing debate. Octyl glucoside (OG) which is among the most commonly used detergents in structural biology [7], was chosen here for purification and crystallisation because of its stabilising impact on PlaF. Notably, PlaF remained in the active and native form during several weeks of crystallisation in the presence of OG detergent. We did not screen other detergents with different hydrophilic-lipophilic balance values and thus cannot provide a statement about the correlation of HLB and PlaF activity, stability and dimerisation. However, in vivo crosslinking experiments showed that PlaF dimers can be formed in the absence of OG (Figure 5a).

3. The mechanism of virulent effect has to be more clearly determined. Does it relate to phospholipase, or caroboxyesterase, or both? Does PlaF induce membrane lysis of macrophages or insect cells by producting lysoGPL?. More mechanical insights are necessary.4. Discussion is necessary how the remodeling enzyme plays also virulent roles.

We have shown that PlaF is associated with the cytoplasmic membrane of *P. aeruginosa* cells; hence, it cannot come into direct contact with the host cells. Therefore, we suggest that PlaF-mediated virulence is not related to the simple lysis of host cells during infections caused by *P. aeruginosa*.

Although PlaF shows PLA and esterase activities, its cellular localisation and lipidomics results suggest that GPLs are native substrates of PlaF. In our view, PlaF regulates the virulence of *P. aeruginosa* by modulating the phospholipid composition of the cytoplasmic membrane with which it is associated. Although we do not entirely understand how these changes of GPL content affect the physiology of *P. aeruginosa*, our global proteomic analysis of wild-type and Δ*plaF* strains revealed that cells respond to the absence of PlaF by changing the production rates of many virulence-related proteins. This notion is in accordance with the widely accepted assumption that the function and structure of membrane proteins (e.g. two-component sensors, transporters) are regulated through protein-lipid interactions. Our preliminary proteomics results agree with these suggestions (for more details, see the answer to the third question of reviewer 1).

We have now discussed the possible role of PlaF-mediated remodelling of GPL on the regulation of virulence proteins in *P. aeruginosa* (page 13, lanes 1-3).

References

1. Kuhle, K., et al., Oligomerization inhibits Legionella pneumophila PlaB phospholipase A activity. J Biol Chem, 2014. 289(27): p. 18657-66.

2. Schunder, E., et al., Phospholipase PlaB is a new virulence factor of Legionella pneumophila. Int J Med Microbiol, 2010. 300(5): p. 313-23.

3. Kovacic, F., et al., A membrane-bound esterase PA2949 from Pseudomonas aeruginosa is expressed and purified from Escherichia coli. FEBS Open Bio, 2016. 6(5): p. 484-93.

4. Ahmad, S., et al., Substrate Access Mechanism in a Novel Membrane-Bound Phospholipase A of Pseudomonas aeruginosa Concordant with Specificity and Regioselectivity. J Chem Inf Model, 2021. 61(11): p. 5626-5643.

5. Fang, L., et al., Genome-scale target identification in Escherichia coli for high-titer production of free fatty acids. Nat Commun, 2021. 12(1): p. 4976.

6. Bocharov, E.V., et al., Helix-helix interactions in membrane domains of bitopic proteins: Specificity and role of lipid environment. Biochim Biophys Acta Biomembr, 2017. 1859(4): p. 561-576.

7. Moraes, I., et al., Membrane protein structure determination – the next generation. Biochim Biophys Acta, 2014. 1838(1 Pt A): p. 78-87.